# Long-read genome sequencing improves detection and functional interpretation of structural and repeat variants in autism

## Graphical abstract

## Authors

Milad Mortazavi, James Guevara, Joshua Diaz, ..., Melissa Gymrek, Abraham A. Palmer, Jonathan Sebat

## Correspondence

jsebat@ucsd.edu

## In brief

Mortazavi et al. investigated structural and tandem repeat variants from long-read WGS in a cohort of 267 individuals from 63 families affected by ASD. They detected *de novo* and complex DUP-DEL SVs, investigated the effect of SVs on imprinted genes and *FMR1*, and performed burden associations and heritability calculations of rare variants.

## Highlights

- Long-read WGS boosted SV/TR detection by 33%/38% and found novel *de novo* SVs

- A previously undescribed class of DUP-DEL complex structural variants was identified

- Joint SV/TR/methylation analysis revealed variants in imprinted genes and *FMR1*

- Rare variants (SVs, TRs, damaging SNVs) explain between 3% and 17% of ASD heritability

 Mortazavi et al., 2026, Cell Genomics 6, 101186
May 13, 2026 © 2026 The Author(s). Published by Elsevier Inc.

CellPress

## Article

# Long-read genome sequencing improves detection and functional interpretation of structural and repeat variants in autism

Milad Mortazavi,[1] James Guevara,[1] Joshua Diaz,[1] Stephen Tran,[1] Helyaneh Ziaei Jam,[2] Chloe Reeves,[3] Sergey Batalov,[4] Kristen Jepsen,[3] Matthew Bainbridge,[4,5] Aaron D. Besterman,[1,4,6,7] Melissa Gymrek,[2,8] Abraham A. Palmer,[1,3] and Jonathan Sebat[1,3,9,10,*]

[1]Department of Psychiatry, University of California, San Diego, La Jolla, CA 92093, USA
[2]Department of Computer Science and Engineering, University of California, San Diego, La Jolla, CA 92093, USA
[3]Institute for Genomic Medicine, University of California, San Diego, La Jolla, CA 92093, USA
[4]Rady Children's Institute for Genomic Medicine, San Diego, CA 92123, USA
[5]Codified Genomics LLC, Houston, TX 77030, USA
[6]Rady Children's Hospital San Diego, San Diego, CA 92123, USA
[7]Laura Rodriguez Research Institute, Family Health Centers of San Diego, San Diego, CA 92101, USA
[8]Department of Medicine, University of California, San Diego, La Jolla, CA 92093, USA
[9]Department of Cellular and Molecular Medicine and Pediatrics, University of California, San Diego, La Jolla, CA 92093, USA
[10]Lead contact
*Correspondence: jsebat@ucsd.edu

## SUMMARY

Long-read whole-genome sequencing (LR-WGS) technologies enhance the discovery of structural variants (SVs) and tandem repeats (TRs). We performed LR-WGS on 267 individuals from 63 autism spectrum disorder (ASD) families and generated an integrated call set combining long- and short-read data. LR-WGS increased detection of gene-disrupting SVs and TRs by 33% and 38%, respectively, and enabled identification of novel exonic *de novo* germline and somatic SVs. We observed complex SV patterns, including a class of nested duplication-deletion events. By joint analysis of phased genetic variation and DNA methylation, we identified deletions of imprinted genes and demonstrated the effect of intermediate TR expansions (35–54 CGG) on the methylation of *FMR1* promoter. Rare SVs, TRs, and damaging SNVs together accounted for 7.4% (95% confidence interval [CI], 2.7%–17%) of the heritability of ASD. These findings demonstrate how LR-WGS can resolve complex genetic variation and its functional consequences and regulatory effects in a single assay.

## INTRODUCTION

Our current understanding of the genetic and neurobiological basis of autism spectrum disorders (ASDs) stems from the identification of ASD susceptibility genes through large-scale studies of rare genetic variants. These discoveries have been driven by advances in genomic technologies applied to large cohorts of parent-child trio families. Investigations of *de novo* copy-number variants (CNVs)[1–3] and single-nucleotide variants (SNVs)[4–6] have led to the identification of over 100 ASD-associated genes, many of which are involved in chromatin regulation, transcriptional control, and synaptic function.[2,7] Despite these significant advances, a substantial portion of the genetic architecture of ASD remains unexplained. Previous studies by our group and others have shown that rare variants account for approximately 4% of the variance in ASD case status, with rare SNVs[8] and CNVs[9] each accounting for around 2%. In comparison, common single-nucleotide polymorphisms (SNPs) are estimated to explain approximately 11% of ASD heritability.[10]

A portion of the genetic contribution to ASD could lie within genomic variation that is poorly captured by current short-read whole-genome sequencing (SR-WGS), particularly rare structural variants (SVs) greater than 50 base pairs (bp) in length[11] and variable number tandem repeats (VNTRs) that cannot be fully resolved with short reads.[12] LR-WGS has demonstrated substantial advantages over SR-WGS in the detection of SVs and TRs. The longer read lengths, ranging in size from 5 to 500 kb,[13] enable direct resolution of repetitive and structurally complex regions that are often collapsed or misassembled in SR-WGS[14,15] and allow the assembly of phased SV haplotypes.[16] LR-WGS technologies have been instrumental in generating telomere-to-telomere (T2T) genome assemblies that have closed long-standing gaps in the human reference genome GRCh38.[17,18] These advances refine our understanding of genome architecture and also enhance the interpretation of pathogenic variants in previously uncharacterized loci.

In addition to improved variant detection and assembly, LR-WGS can accurately call base modifications, particularly DNA methylation, without requiring separate bisulfite or enzymatic

conversion assays. This enables the simultaneous profiling of genetic and epigenetic variation from the same genome sequence, facilitating integrative analyses that link rare variants to methylation episignatures and disease mechanisms.[19–21] Collectively, these features enable more comprehensive genome analysis and variant interpretation.

Through long-read sequencing of 267 genomes, we demonstrate that LR-WGS significantly enhances the detection of SVs and enables detailed characterization of their structure, functional impact, and associated DNA methylation patterns. The comprehensive capabilities of this technology support not only the discovery of novel genetic variation but also their functional characterization.

## RESULTS

### Calling SVs and TRs from LR-WGS

We sequenced 267 individuals (Table S1), 243 of whom are within complete trios from 63 families (some families have more than one offspring). This includes 117 offspring (76 cases, 41 unaffected controls, 74 males, 43 females), and 126 parents. Of the total, 158 individuals were sequenced using the PacBio high-fidelity (HiFi) platform (Sequel IIe) and 109 using Oxford Nanopore Technologies (ONT) platform (GridION), with mean read length of ~5,600 bp for ONT and 11,300 bp for HiFi sequencing (Figure S1). Standard SV and tandem-repeat (TR) calling pipelines were applied to both platforms (see STAR Methods). For SVs highly overlapping with TR regions (>50% reciprocal overlap), the SV call almost invariably represents an expansion or contraction of the TR. For these variants, therefore, we rely on the TR genotyper LongTR,[22] and TR-intersecting SV calls were excluded from the SV call set (non-TR SVs; Figure 1). The resulting SV calls were then merged with existing short-read WGS SV call sets from a prior study with the same subjects.[23]

To benchmark our SV calling workflow, we deeply sequenced one individual (REACH000236) using both PacBio HiFi and Oxford Nanopore platforms. A high-confidence SV truth set was generated by identifying variants concordantly detected by both platforms. We then downsampled the sequencing data to coverages ranging from $2\times$ to $40\times$ and evaluated SV calling performance at each coverage level. Based on these results, we determined that a sample-quality (SQ) threshold of $\geq20$ provided optimal sensitivity and false-discovery rates across coverages (Figure S2). Applying this threshold, the filtered SV call set (excluding TRs) comprised 44,647 alleles, including 22,033 deletions, 19,579 insertions, 2,370 duplications, and 665 inversions. The concordance of the HiFi and ONT platforms for the SV and TR call sets of REACH000236 is evaluated in Figure S3. The concordance for both SV and TR call sets was 88% using our SV and TR detection pipelines.

As expected, the number of SVs detected per sample was positively correlated with sequencing coverage (Figure S4), and coverage was therefore included as a covariate in all burden analyses (STAR Methods). TR regions were genotyped using LongTR.[22] These regions were defined based on the "Simple Repeats" and "RepeatMasker" tracks from the University of California, Santa Cruz (UCSC) Genome Browser, resulting in

918,557 regions genome-wide. Given that structural variation was the primary focus of this study, we applied a threshold of $\geq50$-bp deviation from the reference for TR regions, consistent with the standard definition of SVs. The number of TR regions per sample with length deviations greater than 50 bp relative to the reference was also dependent on sequencing coverage (Figure S4). For each TR region, we calculated a $Z$ score for each haplotype based on the deviation from the cohort-wide distribution of base-pair deviations. These $Z$ scores were used to identify large-deviation outlier haplotypes for burden test analyses (STAR Methods).

### Comparison of SVs and TRs between LR-WGS and SR-WGS

We compared the SV call set generated from LR-WGS to our previously published SV call set from SR-WGS on the same cohort.[8] We also assessed the yield of protein-coding SVs in constrained genes, a functional category previously shown to be strongly associated with ASD susceptibility.[23,24] Constrained genes are defined as genes with probability of loss-of-function intolerance (pLI) $\geq$ 0.9, and ASD genes are defined as genes with supporting evidence from the SFARI gene database (SFARI genes)[25] (Table S2). The number of novel SVs detected exclusively by LR-WGS (16,488) exceeded those detected only by SR-WGS (7,084) (Figure 1A), underscoring the enhanced sensitivity of LR-WGS. However, the substantial number of SR-WGS-specific calls highlights the complementary nature of the two technologies. This pattern was also observed for coding SVs overall (Figure 1B) and for SVs within constrained genes (Figure 1C), where a notable fraction of SVs were unique to each platform (Table S3). Approximately 25% of the SVs in constrained genes were LR-WGS specific. This is a smaller proportion relative to the genome as a whole due to a shift in the SV length distributions across these categories (Figures 1D–1F). Closer manual inspection of constrained-coding variants detected by LR-WGS showed that 90% of these can be verified by the orthogonal SR-WGS data (Table S3).

Short tandem repeats (STRs), defined by motif lengths of 1–6 bp, and VNTRs, with motifs longer than 6 bp, together represent a class of genomic variation (TRs) that has been implicated in a variety of complex traits and diseases.[26,27] Dedicated tools have been developed to genotype these regions using short-read sequencing, such as HipSTR[28] for STRs and adVNTR[29] for VNTRs. However, many TR regions remain inaccessible to SR-WGS. For instance, HipSTR is limited to genotyping STRs that can be fully spanned by a single short read, which dramatically restricts the number of genotypable TR regions. LR-WGS overcomes these limitations by enabling full coverage of longer TR regions within individual reads. Furthermore, the ability to efficiently phase long reads allows for accurate haplotype-specific TR genotyping.

The advantages of LR-WGS for TR genotyping were readily evident. Using LongTR,[22] we were able to genotype 98% of the 918,557 TR regions annotated within repeat elements in the GRCh38 reference genome, as these regions were fully spanned by long reads. Approximately half of these genotyped regions do not overlap with STR regions genotypable using short-read methods such as HipSTR[28] (Figure 1G). Figure 1H

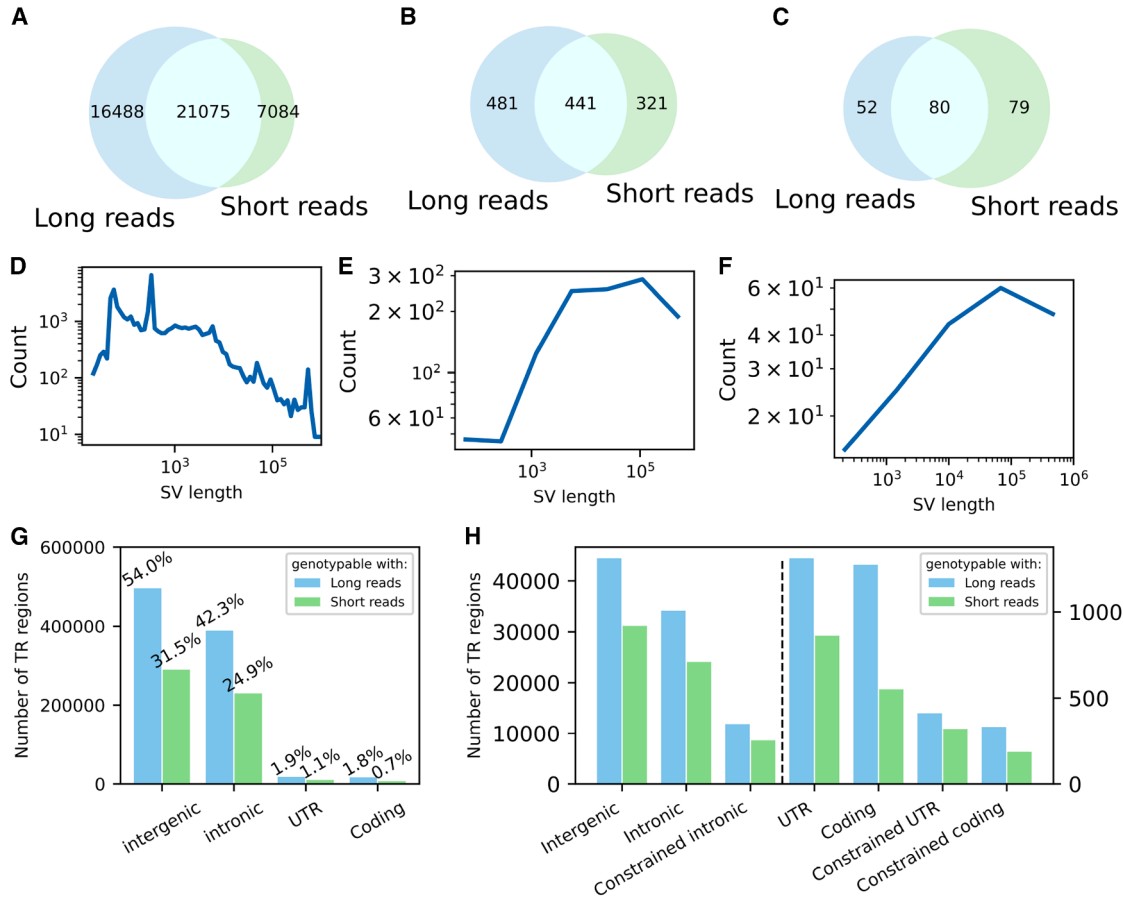

**Figure 1. Contribution of LR-WGS to SV and TR detection**

(A–C) Non-TR SVs (deletions, insertions, duplications, inversions) were filtered using a sample-quality (SQ) threshold of ≥20. Panels show the relative contributions of long-read (LR) and short-read (SR) sequencing platforms to the detection of (A) all non-TR SVs, (B) non-TR SVs intersecting protein-coding regions, and (C) non-TR SVs intersecting coding regions of constrained genes, defined as the union of genes with pLI ≥ 0.9 and the SFARI genes (Table S2). A complete list of non-TR SVs in coding and constrained-coding regions is provided in Table S3.

(D–F) SV length distributions for the same three categories: (D) all non-TR SVs, (E) non-TR SVs intersecting protein-coding regions, and (F) non-TR SVs intersecting constrained-coding regions.

(G) Number of total TRs across genomic regions and the subset that overlap with at least one short tandem repeat (STR) genotypable by HipSTR using SR-WGS. Percentages reflect the proportion of TRs in each category.

(H) Number of TRs in each genomic region for which at least one individual has a haplotype with a base-pair deviation ≥50 bp, genotyping quality >0.9, supported by at least two reads, and with allele frequency <0.5 in parents. These are compared to the subset that intersect STRs genotypable by HipSTR with SR-WGS. Constrained genes are defined as in (C).

A complete list of UTR and coding TR variants is provided in Table S4. See also Tables S2, S3, and S4.

depicts the number of TR regions in which at least one individual harbors a haplotype with a base-pair deviation exceeding 50 bp across various genomic categories (Table S4). The figure also highlights the subset of TR regions that overlap with at least one STR that could be genotyped using SR-WGS, underscoring the enhanced resolution of LR-WGS.

### Novel *de novo* and somatic-mosaic SVs identified by LR-WGS

In genetic studies of SVs in ASD, the strongest genetic association signals come from *de novo* mutations in genes.[1,5] Using LR-WGS, we identified multiple exonic *de novo* SVs (dnSVs) that were not detected in previous SR-WGS analyses of this cohort.[23] Candidate dnSVs were defined as variants supported by ≥3

phased long reads in the proband (allele depth ≥ 3) and homozygous reference genotypes in both parents, each with ≥1 phased long read on both haplotypes. This stringent filtering of long-read SV calls in addition to the SVs that were previously detected from SR-WGS[23] yielded, in total, 65 candidate dnSVs. To validate the putative dnSVs detected from LR-WGS, we performed orthogonal genotyping of putative LR-WGS dnSVs using SR-WGS data (STAR Methods). In total, we confirmed 15 dnSVs, three unique to LR-WGS, six unique to SR-WGS, and six detected by both, in 11 ASD cases and three unaffected controls (one case had two dnSVs) (Table S5).

Three novel dnSVs (detected by LR-WGS and not detected previously by SR-WGS) were detected (Figure 2) in addition to 12 dnSVs previously identified in nine ASD cases and two

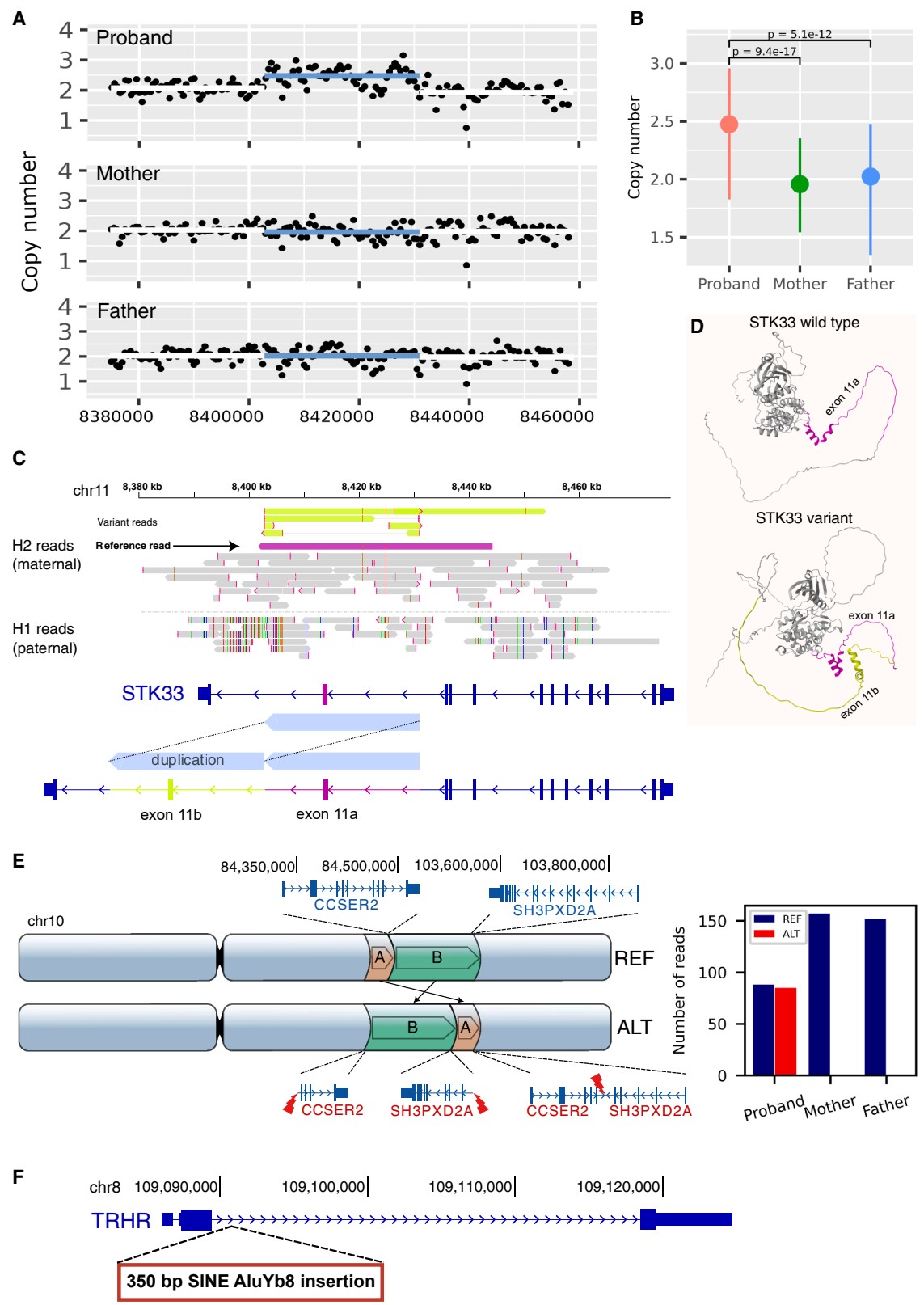

(legend on next page)

controls by SR-WGS.[23] The newly identified dnSVs were in two cases (REACH000426 and REACH000479) and one control (REACH000592), and all occurred within genes. This increased the observed rate of dnSVs in cases from 12% (9/76) to 14% (11/76). One dnSV involved a tandem duplication of the penultimate exon (exon 11) of *STK33*. Copy-number analysis using SR-WGS data revealed an estimated copy number of 2.5 in the proband and 2.0 in both parents, suggesting a heterozygous duplication present in approximately 50% of cells (Figures 2A and 2B). This pattern is consistent with a somatic mutation arising during the first or second cell division post-zygotically. Further support for somatic mosaicism was evident from phased long reads, which showed both alternative (ALT) (yellow) and reference (REF) (magenta) alleles on the maternal H2 haplotype (Figure 2C). Splicing of the duplicated exon was predicted to result in an in-frame duplication of 66 amino acids, encompassing a C-terminal α helix domain and adjacent disordered region of *STK33*. Structural modeling using AlphaFold[30] suggests a substantial alteration to the C-terminal structure of the mutant protein (Figures 2D; S5). The duplicated protein domains encoded by exon 11a (magenta; Figure 2D) and exon 11b (yellow; Figure 2D) are located near the C terminus and are predicted to form a loop in which the helical domains make direct contact with each other. Attempts to validate the mutant transcript by polymerase chain reaction (PCR) amplification from peripheral blood were unsuccessful for both ALT and REF mRNA isoforms due to very low expression of *STK33* in blood (0.01 TPM in GTEx) (Figure S6; see STAR Methods).

Additional dnSVs identified by LR-WGS included a large balanced rearrangement on chromosome 10, spanning approximately 20 Mb (Figure 2E). This rearrangement resulted in the truncation of two protein-coding genes: *CCSER2* (pLI = 0.07) and *SH3PXD2A* (pLI = 1). The *de novo* rearrangement was validated using breakpoint evidence from SR-WGS data in the proband and both parents (sub-panel in Figure 2E). *SH3PXD2A* (Tks5) is an essential gene[32] that is highly intolerant to loss-of-function (LoF) variants (pLI = 1). *SH3PXD2A* is a scaffold protein involved in actin-cytoskeleton remodeling[33] and reactive oxygen species (ROS) signaling,[34] pathways that have been implicated in neurodevelopmental disorders.[35,36]

In another ASD case, we detected a *de novo Alu* element insertion of approximately 350 bp within an intronic region of *TRHR* on the paternal haplotype (chr8:109,090,902) (Figure 2F). While the functional and clinical significance of this variant remains uncertain, biallelic mutations in *TRHR* have been associated with autosomal recessive traits (OMIM #618573).[37]

## Nested duplication-deletion rearrangements are a common form of complex SV with diverse functional consequences

Sequence-level characterization of SVs using LR-WGS reveals substantial genomic complexity.[16,38] In this study, we identified three complex rearrangements characterized by a duplication event followed by a nested deletion within one or spanning both of the duplicated copies. These rearrangements share key features with a complex duplication-deletion (DUP-DEL) rearrangement we recently described in a clinical case report,[39] suggesting that DUP-DEL events may represent a recurrent class of complex genomic rearrangement (Table S8). One example involves a large *de novo* rearrangement in the 8p23.1 region spanning approximately 4 Mb and flanked by two segmental duplications in an individual with a diagnosis of ASD and aggression. LR-WGS resolved this event as an inverted duplication (INV-DUP) of 3.8 Mb between breakpoints A and D followed by a deletion at the junction between the two copies (Figures 3A; S7A). The inverted duplication is evident from the junction sequence of the breakpoints B to C. A schematic illustration demonstrates how this INV-DUP-DEL rearrangement results in a coverage profile resembling a staircase (Figure 3B). In a second case (Figures 3C; S7B), found in a sibling control, a tandem duplication (TAN-DUP) encompasses the full-length isoforms of *ZMYM2* (pLI = 0.96), *GJA3* (pLI = 0.13), and *GJB2* (pLI = 0) as well as the first two exons of *ZMYM5* (pLI = 0). The accompanying deletion affects one of the *ZMYM2* copies. The TAN-DUP-DEL architecture generates a characteristic sawtooth-like coverage pattern, illustrated in Figure 3D. The third example, found in an individual with a diagnosis of ASD and intellectual disability, involves a TAN-DUP within the gene *CDC42BPA* (pLI = 1), accompanied by a partial deletion of one

**Figure 2. Novel *de novo* and mosaic SVs undetected by SR-WGS**

(A) SR-WGS coverage in the duplication region (*chr11:8,402,807–8,430,981*) for subject REACH000426 confirms absence of the variant in both parents. The copy-number increase in the proband does not reach a full additional copy, consistent with mosaic duplication. Coverage was computed in 500 bp windows with mosdepth[31] (see STAR Methods).

(B) Estimated copy number across the duplication region with 95% confidence intervals indicates a *de novo* duplication in the proband. The observed copy number of 2.5 supports a mosaic event, with the variant present in approximately half of cells. Two-sided *t* tests were used to assess statistical significance of the copy-number differences. The number of observations across the duplication region is *n* = 55.

(C) LR-WGS resolves the duplication breakpoints and reveals that the variant resides on the maternal haplotype (H2). Phased reads supporting the duplication are shown in yellow. A ~42-kb-long maternal read spanning both breakpoints supports the reference allele, confirming somatic mosaicism. Duplication of exon 11 of *STK33* is illustrated for the Gencode transcript ENST00000447869.5 (RefSeq NM_001352399.2).

(D) The in-frame duplication of exon 11 of *STK33* is predicted to be structurally tolerated and would not cause a frameshift. AlphaFold models of wild-type and variant proteins are shown. In the variant, the duplicated exon (yellow, 66 amino acids) folds onto the original helical domain (magenta) near the C terminus.

(E) A large *de novo* rearrangement involving two segments, 767 kb (*chr10:83,694,180–84,461,695*, orange) and 19 Mb (*chr10:84,461,666–103,696,005*, green), disrupts *CCSER2* and *SH3PXD2A* in subject REACH000592. The sum of read support for reference and variant alleles from SR-WGS at the three breakpoints confirms the variant as *de novo* (Table S6).

(F) A *de novo Alu* insertion (350 bp) in the intron of *TRHR* at *chr8:109,090,902* in subject REACH000479 was detected on the paternal haplotype (H1). Assembly of paternal reads yielded a consensus sequence consistent with a SINE AluYb8 element (SINE: short interspersed nuclear element) (Table S7).

SV calls described in this figure are listed in Table S5. See also Figures S5 and S6 and Tables S5, S6, and S7.

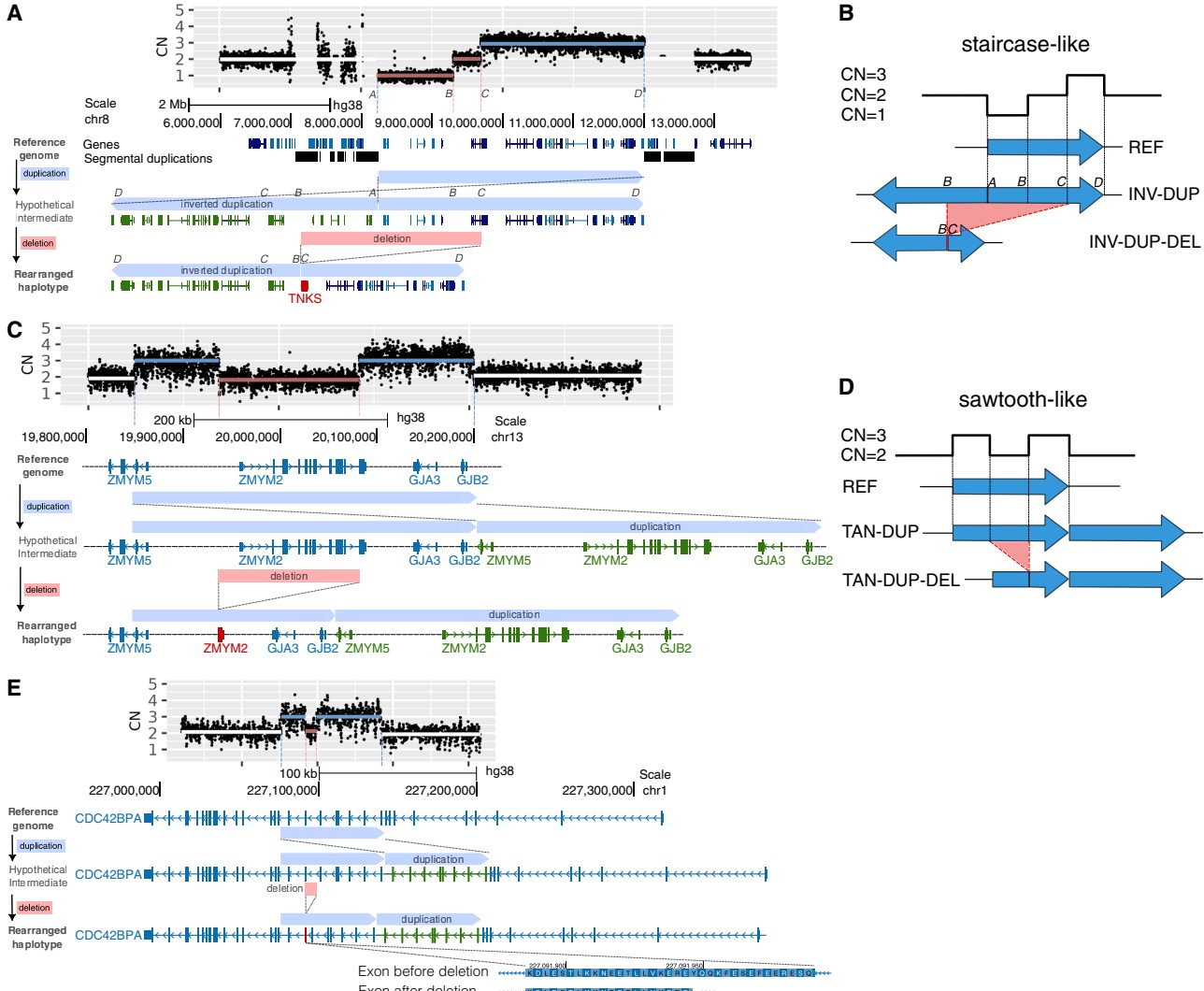

**Figure 3. Complex DUP-DEL SVs exhibit diverse functional effects on genes**

Each panel illustrates a complex DUP-DEL rearrangement in which a duplication (DUP) and deletion (DEL) occur sequentially on the same haplotype. Copied genes/exons are shown in green and genes that are disrupted by the deletion are shown in red. At the top of each panel is the distinct copy-number signature of each DUP-DEL SV from short-read WGS aligned to the reference genome.

(A) A large *de novo* INV-DUP-DEL in subject CLINICAL_S1 involves an inverted duplication (INV-DUP; *chr8:8,200,000–12,000,000*, breakpoints A–D) followed by a deletion that spans the junction between the two copies (*chr8:8,200,000–9,688,994*, breakpoints B–C). The combined rearrangement produces a staircase-like copy-number profile in coverage data from SR-WGS. The orientation of this INV-DUP-DEL could not be determined because there were no DEL-supporting reads that extend beyond the DUP boundary. Either orientation would have the same functional consequence.

(B) Schematic diagram illustrating how the INV-DUP-DEL in (A) produces a staircase-like coverage signature.

(C) A maternally inherited TAN-DUP-DEL (with tandem orientation) in subject REACH000630 includes a duplication (*chr13:19,848,253–20,204,446*) and a nested deletion (*chr13:19,937,353–20,084,249*). This rearrangement results in non-functional remnants of *ZMYM2* and *ZMYM5* on one copy, while the second copy remains intact.

(D) Schematic of a TAN-DUP-DEL rearrangement, in which the deletion is nested within one copy of the DUP, resulting in a characteristic sawtooth-like coverage pattern, as observed in (C) and (E).

(E) A maternally inherited TAN-DUP-DEL in subject REACH000529 consisting of an internal rearrangement of the *CDC42BPA* gene involves a duplication of seven exons (*chr1:227,076,083–227,142,050*) and a nested deletion (*chr1:227,091,947–227,099,486*) that partially deletes one exon. This rearrangement is predicted to result in protein truncation.

For the TAN-DUP-DEL examples that are illustrated in (C) and (E), it is not possible to determine which copy of the DUP contains the DEL because there are no DEL-supporting reads that extend beyond the duplication boundaries. Figure S7 provides more detailed information on the specific signatures of each DUP-DEL rearrangement from the alignments of long reads that span the DUP and DEL breakpoint junctions. Complex SVs described in this figure are listed in Table S8. Additional analysis and results on sequence homologies among the breakpoints of DUP-DEL events is described in the STAR Methods and supplemental information. See also Figures S7, S8, and S9 and Table S8.

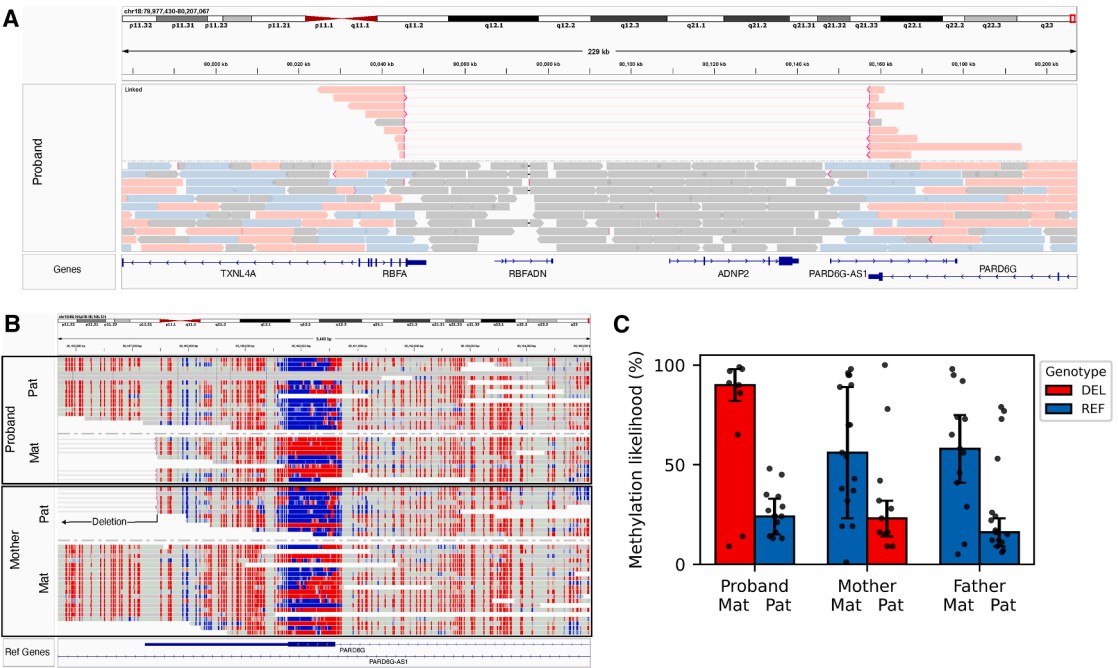

**Figure 4. Deletion of the imprinted gene ADNP2**

(A) A maternally inherited deletion of the maternally expressed gene *ADNP2* was detected in proband REACH000293 (chr18:80,045,344-80,157,432). The deletion spans the full *ADNP2* gene and non-coding segments of the adjacent genes *RBFA* (3′ UTR), *PARD6G* (3′ UTR), and *RBFADN* (lncRNA: long non-coding RNA). Linked reads highlight the breakpoints of the deletion and color represents the haplotype (red, maternal; blue, paternal; gray, unphased).

(B) Methylation of CpG sites (red, methylated; blue, unmethylated) is shown for the imprinting control region. In this case, the methylated (red) haplotype is the maternal (expressed) allele.

(C) Based on phased methylation data, the deletion is on the maternal (expressed) allele in the proband and is present on the paternal (inactive) allele in the mother. The bar heights and error bars correspond to median and median absolute deviation (MAD) of the data.

See also Figure S10 and Table S9.

exon, likely resulting in a truncated protein product (Figures 3E; S7C). Structurally, this rearrangement resembles the TAN-DUP-DEL pattern observed in the previous example (Figure 3D). Alternative genomic mechanisms could result in similar structural configurations; these possibilities are discussed further in Figure S8. The three examples in Figure 3 represent all nested DUP-DEL variants in this cohort. A fourth DUP-DEL signature was detected that was monomorphic in this sample (allele frequency 100%) and may represent a rare ancestral allele or error in the grch38 genome assembly rather than a true common polymorphism in the population (Figure S9). Analysis of the breakpoint sequences of the variants in Figure 3 find evidence for a mixture of mutational mechanisms, including non-allelic homologous recombination (NAHR), micro-homology-mediated end joining (MMEJ), and non-homologous end joining (NHEJ) (supplemental information). In these examples, the duplication and the nested deletion of the same DUP-DEL variant often appear to occur by different repair mechanisms.

## Limited evidence for imprinting disorders

Both LR-WGS platforms used in this study are capable of detecting 5-methylcytosine,[40,41] enabling joint analysis of phased genetic variation and DNA methylation. For instance, phased methylation data from LR-WGS can be used to determine the imprinting status of an allele which can facilitate the diagnosis of an imprinting disorder such as Prader-Willi syndrome (PWS) or Angelman syndrome (AS).[42,43] As expected, several known imprinting control regions (ICRs) showed strongly skewed methylation in our dataset, such as the PWS/AS CNV region (Figures S10A and S10B), as well as *GRB10* (Figure S10C) and *GNAS* (Figure S10D).

We examined the SVs and SNVs detected in our cohort for loss-of-function (LoF) variants in the expressed allele of an imprinted gene, a pattern consistent with an imprinting disorder.[44] Exonic deletions or protein-truncating SNVs were intersected with a database of imprinted genes.[45] LoF variants in four putatively imprinted genes were identified (*ANO1*, *ERAP2*, *ZNF396*, *ADNP2*), of which one gene (*ADNP2*) had an ICR that could be confirmed to have skewed methylation in this dataset. In one subject with ASD, we found a maternally inherited deletion of a maternally expressed gene *ADNP2* (Figure 4A). Skewed methylation of the ICR (chr18:80,159,520-80,160,720) in the trio confirmed that the deletion (chr18:80,045,344-80,157,432) was present on the active (maternal) allele in the proband and on the inactive (paternal) allele in the mother (Figures 4B and 4C). This proband was also determined to have XYY syndrome, a known contributor to autism (Table S5), so, at most, this variant could be a potential genetic modifier.

Figure 5. Expanded gray-zone alleles of *FMR1* are hypermethylated in females

(A) Methylation status of CpG sites near the 5′ UTR of *FMR1* in two female subjects. Reads are grouped by haplotype (H1 and H2), and CpG sites are colored by methylation likelihood: red indicates high methylation, and blue indicates low methylation. The top subject has normal CGG repeat lengths (30 and 29 repeats) and shows random methylation on both haplotypes. In contrast, the bottom subject carries a gray-zone allele (49 repeats) that is fully methylated on one haplotype, while the other haplotype with a normal allele (28 repeats) is fully unmethylated.

(B) Methylation fractions for the long- versus short-CGG haplotypes at *FMR1* in female subjects with ≥3 reads per haplotype. Subjects with at least one gray-zone allele (≥35 repeats) exhibit significant skewing of methylation toward the expanded allele. A logistic regression model classifies subjects based on this skewing, yielding a significant log likelihood *p* value of $7.6 \times 10^{-6}$. Dot sizes reflect the total number of reads on both haplotypes.

*(legend continued on next page)*

The imprinting control region of *ADNP2* is maternally methylated[46] and the *ADNP2* gene is maternally expressed.[45] Activity-Dependent Neuroprotective Protein 2 (*ADNP2*) encodes a homeobox-containing protein expressed in the brain and predicted to act in transcriptional regulation and neuronal function.[47] It is a paralog of *ADNP*, a gene that is associated with ASD and neurodevelopmental disorders.[48] A recent genetic study found weak evidence for association of *ADNP2* with developmental delay but did not detect association with ASD (see Fu et al. Tables S5 and S11[4]). Further investigation of LoF variants in the SPARK (Simons Foundation Powering Autism Research for Knowledge) dataset did not find evidence implicating *ADNP2* or the adjacent maternally expressed gene *PARD6G* (Table S9). This example highlights how joint analysis of phased SVs and methylation can be used to find signatures consistent with an imprinting disorder, but further studies are needed to determine what human traits might be associated with *ADNP2* LoF.

## Expanded gray-zone alleles of *FMR1* are hypermethylated independently of X chromosome inactivation

Fragile X syndrome (FXS), the most common inherited cause of intellectual disability and autism,[49] is caused by *de novo* CGG repeat expansions in the 5′ untranslated region (UTR) of the X-linked gene *FMR1* (chrX:147,912,051–147,912,110).[50] Expansions exceeding 200 CGG repeats result in promoter hypermethylation and silencing of *FMR1*.

Investigation of the activation state of *FMR1* in ASD is another novel analysis that is enabled by LR-WGS. Using data on phased TRs and DNA methylation, we investigated the relationship between *FMR1* CGG repeat length, DNA methylation, and autism case status. Among males (CGG repeat range: 18–41), all *FMR1* alleles were unmethylated (Figures S11A and S11B), consistent with the absence of X chromosome inactivation (XCI). In females, where XCI leads to methylation of most promoter CpG sites,[51] we inferred "activation status"[52] of each long read based on average rate of methylation spanning the CGG repeat (Figure 5A). In a female control (REACH00365) with two average-length alleles (30 and 29 repeats), reads were randomly methylated across haplotypes (binomial test $p$ = 0.55), as expected. In contrast, another female control (REACH00561) carrying a gray-zone allele (49 repeats), defined as 35–54 CGG repeats, showed complete skewing: all reads from the H1 haplotype (28 repeats) were unmethylated, while all reads from the H2 haplotype (49 repeats) were methylated

(binomial test $p$ = 0.0020), indicating allele-specific methylation associated with repeat expansion.

We extended this analysis to all female subjects with at least three phased reads per haplotype and categorized them into three groups based on CGG repeat length: (1) those with at least one gray-zone allele ($\geq$35 repeats, $N$ = 5), (2) those with at least one short allele ($\leq$25 repeats, $N$ = 6), and (3) those with both alleles in the intermediate range (26–34 repeats, $N$ = 11). Among individuals with gray-zone alleles ($N$ = 5), we observed significant skewing of DNA methylation toward the long allele (log likelihood ratio test, $p$ = $7.6 \times 10^{-6}$; Figure 5B). No significant skewing was observed in the other two groups.

To assess whether the observed *FMR1* methylation skewing reflected global XCI, we performed end-to-end phasing of chromosome X using trio-based long-read phasing with WhatsHap and quantified promoter methylation across 163 genes (Tables S10, S11, and S12). In one subject (REACH000479), *FMR1* methylation skewing coincided with global skewing of XCI (Figure 5C) and was associated with a *de novo* truncating variant in *DDX3X*,[8] a gene implicated in an X-linked dominant disorder known to cause XCI skewing.[53] In this subject, skewing was evident for chrX and *FMR1* but not for *DDX3X*, which is a gene that escapes XCI (Figure 5D). In contrast, another subject (REACH00561) exhibited skewed methylation of *FMR1*, but promoter methylation across the remainder of chrX was random (Figure 5E).

Excluding subject REACH000479, CGG repeat length was not correlated with global XCI ($p$ = 0.33; Figure 5F), but showed a significant positive correlation with *FMR1* methylation levels ($p$ = 0.001; Figure 5G). In a combined regression model, CGG repeat length and XCI explained 23% and 15% of the variance in *FMR1* methylation, respectively (Figures S11C and S11D). To assess the functional impact of these epigenetic changes, we performed RNA-seq in nine individuals, five with gray-zone alleles and four controls, and found no significant difference in *FMR1* allelic expression ratios between the two groups ($p$ = 0.7; Figures S11E and S12; Table S13). Additionally, neither CGG repeat length ($p$ = 0.4) nor *FMR1* methylation ratio ($p$ = 0.3) was significantly associated with ASD case status (Table S14). These findings suggest that, although gray-zone alleles are associated with *FMR1* hypermethylation, the functional and clinical significance of this epigenetic effect remains unclear.

## Rare variants in the combined dataset are associated with ASD

Previous studies by our group[1,23,24,54] and others[2,3] have demonstrated that SVs disrupting coding regions of constrained

(C) Comparison of *FMR1* methylation skewness with global XCI skewness across the X chromosome. Most subjects with gray-zone alleles exhibit skewed methylation at *FMR1* but not across the X chromosome, except for REACH000479, who shows global XCI skewing (also shown in D).

(D) Methylation fractions for the two haplotypes across X chromosome genes in subject REACH000479 indicating strong XCI skewing. The red dot marks *FMR1*, and the green dot marks *DDX3X*, where a *de novo* frameshift indel was identified[8] (Table S12).

(E) Methylation profile for subject REACH000561, showing no evidence of global XCI skewing. The red dot represents *FMR1* (Table S12).

(F) Average haplotype methylation across X chromosome genes in female subjects with high coverage (excluding REACH000479), plotted as a function of CGG repeat length. No significant correlation between XCI and CGG repeat length is observed. The $p$ value indicates the significance of the number of CGG repeat length in the linear regression model (Table S11).

(G) *FMR1* haplotype methylation fraction as a function of CGG repeat length, adjusting for XCI (estimated as average methylation across other X chromosome genes), shows a significant positive correlation ($p$ = 0.001; Table S11).

Dot sizes reflect haplotype-specific read counts. The $p$ value indicates the significance of the number of CGG repeat length in the linear regression model. See also Figures S11 and S12 and Tables S10, S11, S12, and S13.

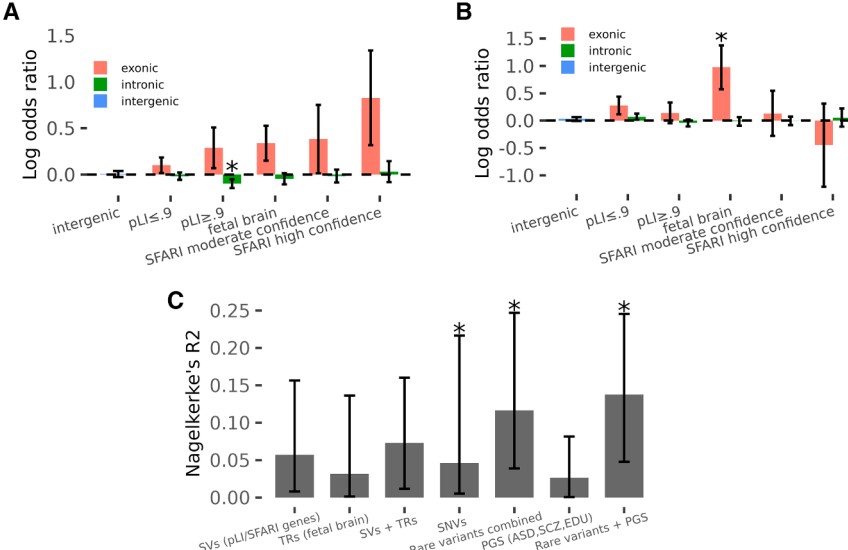

**Figure 6. Association of SV and TR burden with ASD**

(A) Burden-test analysis of SVs using a conditional logistic regression model. SVs included in the analysis had sample quality (SQ) $\geq$20 and population frequency <0.05 (Table S16).

(B) Burden-test analysis of non-homopolymer TRs. Burden scores were computed from high-quality variants (genotype quality >0.9) that showed large repeat length deviations ($\geq$50 bp), high absolute $Z$ scores ($|Z| > 3$), and support from at least two reads (Table S17).

(C) Variance in ASD case status explained ($R^2$) by each variant category. SNVs include *de novo* and inherited loss-of-function (LoF) variants and *de novo* missense variants. Rare variants include SVs, TRs, and SNVs.

Full association results and $R^2$ estimates are provided in Table S18. Asterisks indicate statistical significance ($p$ between 0.01 and 0.05), and error bars represent 95% confidence intervals. See also Figures S13, S14, and S15 and Tables S16, S17, S18, and S19.

genes contribute to autism risk, collectively accounting for approximately 3% of ASD heritability.[9] LR-WGS technologies could expand these studies to a broader range of SVs. This study provides an opportunity to quantify genetic contributions in a dataset of 267 long-read genomes (117 complete trios), but statistical power is limited and larger sample sizes will be required to refine these estimates (Table S15).

Association was investigated for SVs and TRs that intersect with protein-coding exons of genes. Family-based association was tested by conditional logistic regression controlling for coverage, genome-wide SV burden, and additional genetic covariates obtained from our published SR-WGS dataset,[8] including ancestry principal components (PCs; Figure S13), polygenic risk for autism (ASD PRS), burden of *de novo* LoF (dnLoF) and missense (dnMIS), and inherited LoF (inhLoF) SNVs. Observed associations of SVs were directionally consistent with the expectations (Figure 6A; Table S16); however, statistical support in this sample size was modest for SVs affecting constrained (pLI > 0.9) genes ($p = 0.19$), genes highly expressed in the fetal brain ($p = 0.07$), and genes previously implicated in ASD[25] (SFARI genes, $p = 0.08$), and no association was observed for intergenic variants. Association signals were concentrated in large and exonic SVs (Figure S14), particularly in deletions (Figure S15). We also evaluated the burden of rare TR insertions and deletions (length of $|Z| > 3$) excluding homopolymers and low-quality calls. A weak association was observed for exonic TRs in genes expressed in the fetal brain ($p = 0.01$), while no significant signal was detected in other functional categories (Figure 6B; Table S17).

We estimated the conditional contribution of rare variants detected with both LR-WGS and SR-WGS using a joint regression model (Figure 6C; Table S18). Based on partial $R^2$ estimates from the full model, rare SNVs explained 4.6% of the variance ($p = 0.01$), SVs explained 5.7% ($p = 0.054$), and TRs explained 3.2% ($p = 0.13$). Together, rare variants accounted for 11.7% of the variance in case status ($p = 0.012$; 95% confidence interval [CI], 4%–25%), corresponding to 7.4% of the heritability on the liability scale. The total variance explained including polygenic

scores was 13.8% ($p = 0.02$; 95% CI, 5%–25%), corresponding to 8.9% of the heritability (Table S18).

## DISCUSSION

This study leverages LR-WGS to enhance the discovery of structural and repeat variants contributing to ASD. By sequencing 267 individuals from 63 families and directly comparing long-read to short-read sequencing, we demonstrate that LR-WGS substantially improves the detection of gene-disrupting SVs and TRs, particularly those at smaller scales (<1,000 bp) that are often missed by SR-based methods. While SR-WGS remains more sensitive for detecting large coding SVs, owing to the higher number of independent reads contributing to coverage-based signals, LR-WGS offers distinct advantages with respect to determining the functional consequences of SVs. Long reads provide precise resolution of fine-scale structural features and complex rearrangements. Phase information also facilitates detection of somatic mosaicism. Joint analysis of phased genetic variants and DNA methylation enables functional characterization of variants in *FMR1* and imprinted genes.

We identified over 44,000 SVs, approximately 60% of which were novel compared to those detected by SR-WGS. While the majority of coding SVs were captured by SR-WGS, LR-WGS enabled the detection of a broader spectrum of small and complex SVs that were previously unresolved. In addition, LR-WGS identified approximately 11,500 TR variants per subject with length deviations of $\geq$50 bp, more than twice the number detectable using SR-WGS.

We identified *de novo* SVs that were not detected in previous analyses of this cohort.[23] For example, a *de novo* duplication was predicted to result in an in-frame duplication of a helical domain within *STK33*. Phased long-read sequencing confirmed that this duplication occurred on the maternal haplotype and was mosaic in the proband, as evidenced by the presence of both reference and alternate alleles on the same maternal haplotype. This case highlights how phased long reads facilitate precise

characterization of *de novo* variants, including the detection of somatic mosaicism, thereby improving the identification of *de novo* mutations in offspring. This finding contributes to a growing number of *de novo* SVs in this cohort that appear to originate somatically during embryonic development of the parent or the offspring.[24,55]

Sequence-level characterization of SVs revealed previously unrecognized complexity. We identified a recurrent pattern of nested DUP-DEL rearrangements, which also produce distinctive signatures in SR-WGS coverage profiles. One notable example is a DUP-DEL event resulting in LoF of CDC42 Binding Protein Kinase Alpha (*CDC42BPA*), a finding of particular interest given that haploinsufficiency of its paralog, *CDC42BPB*, has been associated with autistic features.[56] Both *CDC42BPA* and *CDC42BPB* are predominantly expressed in the brain and function as downstream effectors of the Rho GTPase CDC42, playing critical roles in regulating cytoskeletal dynamics.[57]

Accurate 5-methylcytosine (5mC) base calling from both PacBio and ONT platforms provides yet another layer of functional characterization that is enabled by long-read sequencing. Joint analysis of phased SVs and DNA methylation identified LoF variants in imprinted genes such as *ADNP2*. Further studies of LR-WGS in large trio cohorts could further elucidate how genetic variation in imprinted genes may contribute to the neurodevelopmental phenotypes. In addition, joint analysis of phased TRs and DNA methylation demonstrates that CGG repeat length in the *FMR1* promoter influences its methylation independently of X inactivation. These findings suggest that earlier reports of skewed XCI associated with gray-zone alleles[58] may actually reflect a *cis*-regulatory effect of CGG repeat length rather than a skewing of XCI.

The burden of rare exonic SVs, TRs, and SNVs in genes explained approximately 7.6% of the heritability of ASD, with partial contributions of 3.5% for SVs, 1.9% for TRs, and 2.8% for rare SNVs. While a sample size of 267 long-read genomes is large by current standards, this dataset is underpowered to detect associations with specific functional categories or genes. Application of LR-WGS to larger cohorts is needed to identify novel associations and to refine our estimates of the heritability explained by SVs and TRs.

In summary, LR-WGS uncovers substantial previously hidden variation, particularly complex SVs and TRs with regulatory or coding consequences. This approach enables base-level phasing, precise variant annotation, and direct methylation profiling, providing a more comprehensive view of the genome's functional architecture. Although the current study is limited by a modest sample size, it demonstrates the utility of long-read sequencing platforms for ASD gene discovery and highlights multiple mechanisms by which SVs and TRs may influence phenotypic outcomes. Future studies involving larger cohorts and deeper sequencing coverage will further improve heritability estimates and refine variant interpretation, thereby advancing our understanding of the genetic architecture of ASD.

### Limitations of the study

The most significant hurdles that we face in the application of new sequencing technologies to clinical cohorts are limitations in sample size and statistical power. During the data collection phase of this project, sequencing of very large samples with the ONT GridION and PacBio Sequel II platforms was cost-prohibitive. The tradeoff between sequencing coverage and sample size was optimized by benchmarking and evaluating performance of SV and TR calling at varying levels of coverage. Sequencing at reduced (4–10×) coverage enabled us to achieve a sample size of 267 subjects. While still underpowered, the sample size was sufficient to functionally characterize clinically relevant rare protein-coding variation and to obtain estimates of the variance explained by the combined measures of rare variant burden (SNVs, TRs, and SVs). Variability attributable to sequencing technologies and coverage was addressed by platform-stratified meta-analysis that controlled for platform, coverage, and genome-wide rare-variant burden. Other limitations include the use of peripheral blood samples for functional characterization of variants. Patterns of methylation and gene expression in peripheral blood may not accurately reflect the functional impact of SVs and TRs in the brain. In some genes, such as *STK33*, mRNAs were undetectable in blood. The results presented here demonstrate new directions for future work using long-read WGS to characterize genetic contributors to neurodevelopmental conditions when phased single-nucleotide, structural, and repeat variation and methylation is obtained from a single assay.

### RESOURCE AVAILABILITY

#### Lead contact

Requests for further information and resources should be directed to and will be fulfilled by the lead contact, Jonathan Sebat (jsebat@ucsd.edu).

#### Materials availability

This study does not generate new unique reagents.

#### Data and code availability

- The aligned sequencing data (bam files) and variant data (VCF files) have been deposited at the NIMH Data Archive: https://doi.org/10.15154/qpjh-dk51.
- All original analysis code is available at Zenodo: https://doi.org/10.5281/zenodo.18381260.
- The long-read genotyper, *snoopSV*, is available at Zenodo: https://doi.org/10.5281/zenodo.18381247.
- Any additional information required to reanalyze the data reported in this paper is available from the lead contact upon request.

### ACKNOWLEDGMENTS

The authors would like to acknowledge grants to J.S. from the NIMH (MH113715, MH133899), grants to A.A.P. from the National Institute on Drug Abuse (U01DA051234), and grants to M.G. from the National Human Genome Research Institute (1R01HG010149). We give special thanks to the Beyster Family Foundation and the Donald C. and Elizabeth M. Dickinson Foundation for philanthropic support. We also acknowledge Dr. Fritz Sedlazeck, Dr. Michael C. Schatz, Dr. Flora Tassone, and Claudia MB Carvalho for the constructive discussions.

### AUTHOR CONTRIBUTIONS

J.S. designed the study, coordinated data collection, and supervised data processing and data analysis. M.M. performed data processing; developed the in-house SV genotyper (snoopSV); and performed burden-test analyses, methylation analyses, and other tertiary analyses. J.G. performed initial data processing. J.D. performed Oxford Nanopore long-read sequencing. M.G.

and H.Z.J. developed the TR genotyper (LongTR). S.B., M.B., and A.D.B. provided data for a complex structural variation. S.T. performed qPCR, iSeq 100 sequencing, and the following RNA analysis. A.A.P. provided scientific and technical support on DNA sequencing. C.R. and K.J. performed library preparation and sequencing of PacBio samples. M.M. and J.S. wrote the paper.

## DECLARATION OF INTERESTS

The authors declare no competing interests.

## DECLARATION OF GENERATIVE AI AND AI-ASSISTED TECHNOLOGIES IN THE WRITING PROCESS

During the preparation of this work, the authors used ChatGPT to help revise the original text for length and readability. After using this tool, the authors reviewed and edited the content as needed and take full responsibility for the content of the publication.

## STAR★METHODS

Detailed methods are provided in the online version of this paper and include the following:

- KEY RESOURCES TABLE
- EXPERIMENTAL MODEL AND STUDY PARTICIPANT DETAILS
- METHOD DETAILS
  - Sequencing data generation
  - SV detection and genotyping
  - TR genotyping
  - Evaluation of SV calling accuracy
  - Selection of SV and TR analysis tools
  - Examining effects of coverage on genotyping accuracy
  - Variant annotation
  - Assigning disease status to subjects in the cohort
  - *De novo* detection from LR-WGS and SR-WGS call sets
  - Copy number calculation from SR-WGS
  - Burden test analysis of SVs and TRs
  - Methylation calling and analysis
  - Possible mechanisms of complex SV generation
  - RNA extraction from whole blood samples
  - Reverse transcription of RNA
  - PCR and gel electrophoresis of *STK33* mRNA
  - PCR and iSeq 100 for *FMR1* mRNA
  - Data processing of Iseq 100
  - Analysis of breakpoint sequences of DUP-DEL events
- QUANTIFICATION AND STATISTICAL ANALYSIS

## SUPPLEMENTAL INFORMATION

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

## Article

CellPress

## STAR★METHODS

### KEY RESOURCES TABLE

| REAGENT or RESOURCE | SOURCE | IDENTIFIER |
|---|---|---|
| **Deposited data** | | |
| Aligned long-read sequencing data (bam files) and variant data (VCF files) | This paper | NIMH Data Archive: https://doi.org/10.15154/qpjh-dk51 |
| snoopSV v1.0.0 | This paper | Zendo: https://doi.org/10.5281/zenodo.18381247 |
| Analysis code for this paper | This paper | Zendo: https://doi.org/10.5281/zenodo.18381260 |
| **Software and algorithms** | | |
| Guppy v4.0.11 | Oxford Nanopore Technologies | RRID:SCR_023196 |
| Dorado v0.6.0 | Oxford Nanopore Technologies | RRID:SCR_025883 |
| SMRT Link | Pacific Biosciences Inc. | https://www.pacb.com/support/software-downloads/ |
| Jasmine v2.4.0 | Pacific Biosciences Inc. | https://github.com/PacificBiosciences/jasmine |
| Cramino v1.0.0 | De Coster et al.[59] | https://github.com/wdecoster/cramino?tab=readme-ov-file |
| minimap2 v2.26 | Li[60] | RRID:SCR_018550 |
| Mosdepth v0.3.10 | Pedersen et al.[31] | RRID:SCR_018929 |
| WhatsHap v2.8 | Garg et al.[61] | RRID:SCR_025319 |
| Sniffles2 v2.2 | Smolka et al.[62] | RRID:SCR_017619 |
| LUMPY v0.2.13 | Layer et al.[63] | RRID:SCR_003253 |
| svtools v0.2.0 | Hall lab[64] | Zendo: https://doi.org/10.5281/zenodo.49391 |
| SV2 v1.4.3.4 | Antaki et al.[54] | https://github.com/dantaki/SV2 |
| LongTR v1.0 | Ziaei Jam et al.[22] | https://github.com/gymrek-lab/LongTR |
| Variant Effect Predictor (VEP) | McLaren et al.[65] | RRID:SCR_007931 |
| Slivar v0.3.0 | Pedersen et al.[66] | https://github.com/brentp/slivar |
| Hifiasm v0.25.0 | Cheng et al.[67] | RRID:SCR_021069 |
| HISAT2 v2.2.1 | Kim et al.[68] | RRID:SCR_015530 |
| AlphaFold | Google DeepMind | https://alphafoldserver.com/welcome |
| **Other** | | |
| Zymo Direct-zol RNA Miniprep Plus kit | Zymo | Cat# R2073 |
| DNA/RNA shield solution | Zymo | Cat# R1200-25 |
| proteinase K | Zymo | Cat# D3001-2-5 |
| iScript cDNA synthesis kit | Bio-Rad | Cat# 1708891 |
| Platinum SuperFi II PCR kit | Thermofisher | Cat# 12361010 |
| 6X loading dye | Thermofisher | Cat# R0611 |
| GelGreen | Biotium | Cat# 41005 |
| 100bp DNA ladder | Thermofisher | Cat# SM0241 |
| QIAquick Gel Extraction Kit | Qiagen | Cat# 28706 |
| AMPure XP Beads | Beckman Coulter | Cat# A63880 |
| Qubit dsDNA BR assay | ThermoFisher | Cat# Q32850 |

## EXPERIMENTAL MODEL AND STUDY PARTICIPANT DETAILS

Samples were collected previously as part of our project "Relating Genes to Adolescent and Child Mental Health" (REACH).[24] Individuals were referred from clinical departments at Rady Children's Hospital, including the Autism Discovery Institute, the Departments of Psychiatry, Neurology, and Speech and Occupational Therapy, and the Developmental Evaluation Clinic. Further referrals came directly through our project website. Each child included in the study received a diagnosis of ASD on the basis of an evaluation

by a licensed clinician.[69] Prior to appointments, families were provided with institutional-review-board-approved consent forms and Health Insurance Portability and Accountability consent forms. DNA was obtained from 5 mL blood draws.

## METHOD DETAILS

### Sequencing data generation

Oxford Nanopore sequencing was performed on the GridION platform at the Sebat Lab. Raw signal data (fast5 files) were basecalled using *Guppy* (v4.0.11), and methylation calling was subsequently performed using *Dorado* (v0.6.0) with the model dna_r9.4.1_e8_-hac@v3.3, after converting fast5 to pod5 format. PacBio HiFi sequencing was conducted by the Institute for Genomic Medicine (IGM) and the Salk Institute. A subset of the data initially generated in continuous long-read (CLR) format was converted to HiFi format in the Sebat Lab using *SMRT Link* software. Sequencing coverage, yield, and read length statistics were obtained using *cramino* (v1.0.0)[59] and are summarized in Table S1.

### SV detection and genotyping

Long reads were aligned to the GRCh38 reference genome using *minimap2*,[60] and sequencing coverage was calculated using *mosdepth*.[31] Phasing was performed in a trio-aware setting using *WhatsHap*[61] to phase previously identified SNVs[8] and assign haplotags to long reads. Structural variants (SVs) were detected for each subject using two independent SV callers: *Sniffles2* (v2.2)[62] and *LUMPY* (v0.3.1).[63] For *Sniffles2*, SVs were first called per individual to generate.snf files, which were then merged using *Sniffles2* with lenient parameters (–combine-low-confidence 0 –combine-low-confidence-abs 1 –combine-null-min-coverage 2 –combine-output-filtered). These relaxed settings ensured broad inclusion of candidate SVs, which were subsequently filtered using an in-house SV genotyper. For *LUMPY*, SVs were called per individual using the following parameters: back_distance = 10, min_mapping_-threshold = 20, weight = 1, and min_clip = 20. Individual call sets were then merged using *svtools*.[64] Both *Sniffles2* and *LUMPY* call sets were genotyped using *snoopSV*, an in-house Bayesian genotyping framework developed to detect and classify SV signatures from long reads (available at Zenodo: https://doi.org/10.5281/zenodo.18381247). For each subject and variant, *snoopSV* reports the number of supporting reads for both the reference and alternate alleles, assigns a genotype (0/0, 0/1, 1/1, phased genotypes if haplotag information is available), and provides both a genotype quality (GQ: Phred-scaled probability that the genotype is correct) and a sample quality (SQ: Phred-scaled probability that the genotype is non-reference).

$$p(Data|GT) = \begin{cases} \binom{Var + Ref}{Var}\left(\frac{1}{2}\right)^{Var}\left(\frac{1}{2}\right)^{Ref}, GT = 0/1 \\ \binom{Var + Ref}{Var}(1-e)^{Var}(e)^{Ref}, GT = 1/1 \\ \binom{Var + Ref}{Var}(e)^{Var}(1-e)^{Ref}, GT = 0/0 \end{cases}$$

$$p(Data) = \sum_{GT} p(Data|GT)p(GT)$$

$$p(GT = 0/0) = p(GT = 0/1) = p(GT = 1/1) = \frac{1}{3}$$

$$p(GT|Data) = \frac{p(Data|GT)p(GT)}{p(Data)}$$

$$GT = Argmax_{GT}\, p(GT|Data)$$

$$GQ = -10\, log_{10}(1 - p(GT|Data))$$

$$SQ = -10\, log_{10}(p(GT = 0/0|Data))$$

We first merge the *Sniffles2* and *LUMPY* call sets. Call sets are merged for each SV type separately. For deletions, duplication and inversions which have two breakpoints per SV, we merge the SVs using *bedtools intersect* with 50% reciprocal overlap to detect the SVs which are called by both call sets. For the SVs which are present in both call sets we use the *Sniffles2* breakpoints (due to higher accuracy). Since *Lumpy* does not detect insertions, we use *Sniffles2* insertions for the long-read SV call set. To avoid over-filtering, we apply a more lenient criterion at this stage, requiring at least one supporting read across all subjects for an SV to be retained. Since tandem repeats (TRs) were genotyped separately, we exclude SVs overlapping TR regions, defined by UCSC Table Browser annotations, by more than 50% reciprocal overlap. The resulting high-confidence, non-TR SV call set was then merged with a previously

published short-read SV call set,[23] keeping track of the platform of origin (long-read or short-read) for each SV. Merging the long-read and short-read SV call sets are identical to the aforementioned procedure. Except for insertions, we use SV length to extend the insertion breakpoint in both directions symmetrically by half of the insertion length to define two pseudo breakpoints and similar to what was done for the other SV types, we use *bedtools intersect* to merge the SVs. For SVs which are common in long-read and short-read call sets, we prioritize long-read breakpoints when merging them (due to higher breakpoint-level accuracy of long-read SV calls).

To cross-validate platform-specific calls, we genotyped LR-only SVs using *SV2*[54] on SR-WGS data, and SR-only SVs using *snoopSV* on LR-WGS data. SVs with supportive genotyping evidence on both platforms were reassigned to the intersection set. The impact of different filtering thresholds, based on SQ and ALT allele read depth, is summarized in Figure S16.

### TR genotyping
TR regions were defined by merging the "Simple Repeats" track and the "simple" sub-track of the "RepeatMasker" track from the UCSC Genome Browser, resulting in a total of 918,557 annotated regions. TRs located within 100 bp of each other were merged, and 30 bp flanking regions were added to both ends of each TR interval. Genotyping was performed using *LongTR*[22] (v1.0) with the following parameters: –phased-bam –min-mean-qual 0 –min-mapq 1 –alignment-params −1.0,-0.458675,-1.0,-0.458675,-0.00005800168,-1,-1. TR genotyping was conducted jointly within each family, and resulting genotypes were merged across families. For each TR region, base pair deviations from the reference of the haplotypes in the cohort were used to compute Z-scores. Genotype quality scores (ranging from 0 to 1), were used to identify high-confidence TR variants ($q > 0.9$) for downstream analyses.

### Evaluation of SV calling accuracy
We benchmarked our in-house SV genotyper using a deeply sequenced individual (REACH000236) with $40\times$ Oxford Nanopore (ONT) and $15\times$ PacBio HiFi coverage. A high-confidence non-TR SV truth set was generated by identifying variants supported by at least three reads from both platforms using *Sniffles2* (v2.2).[62] To evaluate performance, we applied the pipeline to downsampled ONT and HiFi data and generated the receiver operating characteristic (ROC) curves (Figure S2). Based on these results, we established a sample quality (SQ) threshold of $\geq 20$ as an appropriate filter for detecting high-quality SVs. To evaluate Concordance of SVs detected by the HiFi and ONT platforms in the high-coverage individual, we measured the overlap between the SV call sets from each platform and observed 88% concordance (Figure S3B). The challenge of calling SVs within TRs with sniffles2 in sample REACH000236 is illustrated by a representative example in Figure S3E. Long TR detects a 225 bp DEL on haplotype H1 and a 56 bp INS on H2, and the genotypes made by LongTR are identical in HiFi and ONT. From the read alignments, several deletions and insertions can be seen that vary significantly between reads. Multiple SV calls are made by sniffles2 within this region, all of which are discordant between platforms.

### Selection of SV and TR analysis tools
During the course of this project multiple SV calling methods were evaluated including sniffles2, pbsv,[70] cuteSV[71] and SVIM.[72] Selection of sniffles2 as the method of choice was primarily based on features that were desirable for a large-scale project on multiple long read platforms. (1) A key advantage of sniffles2 from our perspective was its introduction of the.snf file format, which stores detailed breakpoint and alignment information for each sample. This file format enabled joint genotyping across multiple genomes, producing a unified SV genotype matrix similar in concept to GATK's gVCF workflow for small variants. (2) In addition, sniffles2 provided automatic parameter optimization for both PacBio and ONT platforms. In the early years of this study, sniffles2 was the only method available that had both of these capabilities. While pbsv did offer a joint genotyping feature, parameters were highly tuned for the PacBio platform.

The choice of LongTR for TR genotyping is based on its capability to genotype a general list of pre-defined TR regions genome-wide (STRs and VNTRs) unbiasedly for both platforms. In addition, it is sufficiently efficient to genotype hundreds of thousands of TR regions for a large cohort.

### Examining effects of coverage on genotyping accuracy
While WGS can outperform microarrays even at low coverage,[73] the tradeoffs between sequencing coverage, variant calling accuracy, and sample size is something that requires careful consideration. The following downstream steps were performed to account for the effects of low coverage on results.

1. Optimize downstream quality control (QC) to produce call sets with comparable quality despite differences in the total of variation that may be captured on a given platform.
2. Control for the effects of coverage in statistical models.
3. Stratification by platform and combining platforms by meta-analysis.

### QC
Due to differences in base calling accuracy, the relationship of coverage to performance differs by platform. In this dataset, ONT required twice the coverage (area under the curve [AUC] >0.7 at 5X, AUC >0.8 at 10X) that HiFi required (AUC>0.7 at 2X, AUC>0.8 at 5X) to reach comparable accuracy. (Figure S2). Hence, our ONT dataset (mean coverage 10.7X) was sequenced to more than twice the coverage of the HiFi (mean coverage 3.8X) dataset, and accuracies were similar at the specified threshold of

sample quality (SQ ≥ 20). The median number of variants captured on average differed by ~20% between PacBio (median = 5,058 SVs) and ONT (median = 6,307 SVs) (Figure S4). The effects of platform and coverage were further accounted for as follows.

### Statistical models

Coverage has a significant influence on the burden of variants detected in a sample (Figure S4), thus it is essential to control for coverage in statistical burden tests. Our statistical models test for SV burden within specific functional categories and account for the effects of coverage and the genome-wide burden of SVs detected. (refer to the section about burden test analysis in **Methods**).

### Stratification by platform

As we have shown previously,[9] genotyping platforms are the single biggest confounding variable in genetic association studies of SVs. We have further shown that spurious signals attributable to the platform are addressed by generating association results separately by platform.[9] Here we generated association statistics for ONT and PacBio datasets and combined results by meta-analysis based on a published method METAL.[74]

### Stratification of association statistics by SV size and functional element

Last but not least, spurious results that are driven by low coverage should broadly affect many types of SVs. We would expect that spurious results would similarly affect large and small SVs, and genic and intergenic SVs, and associations would not be concentrated in autism genes. Therefore, we have stratified our association data by functional consequence (intergenic, intronic, exonic) and size (Figure S14). We observe no significant associations for intronic and intergenic SVs across all size ranges, and the only association signal that is evident consists of coding SVs (Figure S14).

### Variant annotation

SVs and TRs are functionally annotated using the Variant Effect Predictor (*VEP*),[65] with GENCODE v42[75] as the gene model and ENCODE[76] annotations for *cis*-regulatory elements (CREs). Genes are annotated with constraint scores, including probability of loss-of-function intolerance (pLI) which were obtained from the Genome Aggregation Database (gnomAD v4: https://gnomad.broadinstitute.org). SFARI genes are also used to annotate variants impacting genes which are previously associated with ASD.[25]

### Assigning disease status to subjects in the cohort

Case status was assigned based on clinical diagnostic reports evaluated for each subject.[23] Individuals diagnosed with autism, developmental delay, Asperger syndrome, or Pervasive Developmental Disorder Not Otherwise Specified (PDD-NOS) were classified as cases. Additionally, all probands were assigned case status, with the exception of two subjects, REACH000450 and REACH000518, who were confirmed as controls.

### De novo detection from LR-WGS and SR-WGS call sets

Variants from the LR-WGS call set were evaluated for *de novo* status in each trio using *Slivar*.[66] High-confidence *de novo* SVs were defined as those with alternative (ALT) allele depth (AD) ≥3 in the offspring, and zero ALT AD in both parents, each of whom must have ≥1 phased long read supporting the reference (REF) allele on both haplotypes. For variants identified exclusively by SR-WGS, *de novo* status was assessed based on the genotypes of the proband and both parents, contingent on the variant being flagged as high-quality (PASS_STRICT) in the original study.[23]

Candidate *de novo* SVs from LR-WGS were validated by confirming the presence of supporting breakpoint evidence in orthogonal SR-WGS data. For deletions and duplications, validation criteria included multiple discordant read pairs and consistent shifts in coverage depth across the SV region. For inversions, clusters of read pairs with discordant orientation served as evidence. For insertions, in addition to discordant read pair signatures, mate pairs aligning to different chromosomes, suggestive of mobile element insertions (e.g., LINEs or SINEs: long/short interspersed nuclear elements), were used as supporting evidence.

To investigate the *de novo* insertion in subject REACH000479 (Figure 2F), we assembled the paternal haplotype using *Hifiasm*[67] and aligned the 350 bp insertion sequence using *BLAST*.[77] The inserted sequence was identified as a SINE AluYb8 element with 98% sequence identity. The full inserted sequence can be found in Table S7. To visualize methylation at the insertion site, we created a merged haplotype for REACH000479 by adding the *de novo* insertion (from the paternal haplotype) to the maternal haplotype sequence. Long reads from all trio members were then aligned to this contig, revealing full methylation of CpG sites within the *de novo Alu* insertion as well as within a nearby common *Alu* element shared by the trio (Figure S17).

### Copy number calculation from SR-WGS

The copy number calculations in Figs. 2 and 3 are done using *mosdepth*[31] to obtain the local coverage from SR-WGS in 100 or 500 bp window sizes. Local GC content is also used for GC correction of local coverages using the *loess* function in R: *loess(coverage ~ GC_content)*. The coverage values are normalized by those of the flanking regions to obtain the copy number values.

### Burden test analysis of SVs and TRs

The SV types included in burden score calculations comprise deletions (DEL), insertions (INS), duplications (DUP), and inversions (INV). SVs were filtered to include only those with a sample quality (SQ) greater than 20 and a population frequency below 0.05 in parents. For SVs identified by long-read sequencing, parental frequency was calculated based on the presence of at least two supporting reads; for short-read–derived SVs, frequency was based on the presence of a non-reference genotype. To reduce false

positives, SVs with breakpoints located within paired segmental duplications were excluded, as these are more likely to arise from mapping artifacts and are associated with elevated error rates. SV burden within the defined functional categories was defined as the total number of intersecting rare (allele frequency <0.05) deletions, insertions, duplications, and inversions (Table S19). Additionally, one subject in the cohort was found to carry an XYY karyotype (REACH000293); this aneuploidy was treated as an additional SV contributing to the burden score in the high pLI and fetal brain gene categories for that individual.

For the burden test analyses of TRs, we excluded homopolymer TRs due to their higher genotyping error rates. We also removed TR regions where more than 25% of subjects lacked genotype calls, typically due to insufficient coverage or other technical limitations. In addition, subjects with more than 50% missing genotypes across TR regions were excluded from the analysis (Figure S18). A burden score was computed for each subject, defined as the number of TR regions meeting all of the following criteria: high genotype quality (genotype quality >0.9), large deviation in length from the reference ($\geq$50 bp), large absolute $Z$ score ($|Z| > 3$), and support from at least two long reads. Individual burden scores for all subjects are provided in Table S19.

The full model including the burden variable is as follows:

Phenotype $\sim$ category_burden + sex + coverage + count_dnlof_inhlof_dnmis + PRS_ASD_Z + genome_wide_burden + PC1 + PC2 + PC3 + PC4 + PC5 + PC6 + PC7 + PC8 + PC9 + PC10 + strata(fid).

The null model with covariates which is used is as follows:

Phenotype $\sim$ sex + coverage + count_dnlof_inhlof_dnmis + PRS_ASD_Z + genome_wide_burden + PC1 + PC2 + PC3 + PC4 + PC5 + PC6 + PC7 + PC8 + PC9 + PC10 + strata(fid).

Using ANOVA to find the significance of the burden variable ($p$p-value):

ano <- anova(null_model, full_model, test = 'LRT').

In the R-squared calculations, the SV category includes the burden of exonic SVs in high-pLI genes, combined with the CNV burden of each subject. CNV burden was estimated by intersecting SVs identified in this study, specifically deletions and duplications of any quality, with previously reported CNV breakpoints, requiring a 50% reciprocal overlap. Subjects with intersecting SVs were considered to carry a CNV. The TR category reflects the burden of exonic TRs in fetal brain–expressed genes, while the SNV category encompasses the burden of *de novo* loss-of-function (dnLOF), *de novo* missense (dnMIS), and inherited loss-of-function (inhLOF) SNVs. Covariates in the model included sex, sequencing coverage, the first 10 ancestry principal components, and genome-wide SV and TR burden. We fit a full model and a corresponding null model, excluding the predictor(s) of interest for each variant class, using conditional logistic regression. The Nagelkerke's R-squared value was computed as a point estimate of variance explained, and the 95% confidence interval was derived via bootstrapping.

The full model for calculating R-squared:

Phenotype $\sim$ count_snvs + count_svs_PLIp9_cds + count_PLIp9_utr + count_cnvs + count_trs_cds + count_trs_utr + PRS_ASD_Z + PRS_SCZ_Z + PRS_EDU_Z + genomewide_svs + genomewide_trs + sex + MEAN_COVERAGE + PC1 + PC2 + PC3 + PC4 + PC5 + PC6 + PC7 + PC8 + PC9 + PC10 + strata(fid).

Using conditional logistic regression model to fit the data:

clogit(model_formula, data = df_main, method = "approximate").

The null model, for each category, is obtained by omitting the variables of interest from the full model. And the Nagelkerke's R-squared is computed as:

loglik0 = logLik(null_model).
loglikM = logLik(full_model).
cs_r2 = 1.0 - exp(2/N * (loglik0 - loglikM)).
ngk_r2 = cs_r2/(1.0 - exp(2/N * (loglik0))).

We confirmed that there are no associations between the genome-wide burden of SV and TR variants with case status, sex or age of the individual in the cohort (Figure S19; Table S20). We have also performed power analyses for SV and TR burden based on permutations (Table S15). We estimate that a sample size of ~500 families are needed to have reasonable power to detect genetic associations for the functional categories of SV and TR analyzed in this study, and a sample ~18% larger is required for a dataset with 4X coverage (such as our PacBio dataset) compared to a dataset with 10X coverage (such as our ONT dataset).

### Methylation calling and analysis

Methylation data for PacBio HiFi reads was added using *jasmine* (v2.4.0) prior to alignment. For Oxford Nanopore (ONT) data, methylation was derived by re-basecalling pod5 files using *dorado* (v0.6.0) with the methylation model dna_r9.4.1_e8_hac@v3.3.

To estimate CGG repeat sizes in the *FMR1* 5′UTR, we used *snoopSV* to count the number of base pair deviations from the reference sequence for each read spanning the repeat region. The CGG repeat size for each haplotype was then computed as the mean number of repeats across all reads assigned to that haplotype. CpG methylation likelihoods within the *FMR1* UTR were found to be highly consistent within individual reads, with most reads exhibiting either fully methylated or fully unmethylated profiles. Accordingly, each read was classified as methylated or unmethylated based on its average CpG methylation likelihood. The haplotype-level methylation fraction was calculated as the proportion of reads classified as methylated for each haplotype.

The fraction of haplotype methylated for X chromosome genes was calculated as the average methylation fraction across all investigated genes on the X chromosome (Table S10), per haplotype. For each gene, the methylation fraction was defined as the proportion of reads assigned to that haplotype that were classified as methylated, and the chromosome-wide value was obtained by

averaging these gene-level fractions. Gene-level methylation skewness was defined as the difference between the methylation fraction of the long CGG repeat haplotype and that of the short CGG repeat haplotype. To assess methylation skewness between haplotypes, a binomial test was applied using the function binom.test(x, n, $p$ = 0.5, alternative = "two.sided"), where $x$ equals the number of methylated reads in haplotype H1 plus the number of unmethylated reads in haplotype H2, and $n$ is the total number of reads across both haplotypes.

We use an ordinary least square model to associate methylation fraction of *FMR1* as a function of CGG repeat size and X chromosome inactivation (Figures 5G; S5C, S5D):

import statsmodels.api as sm
y = np.array(df.frac_hap_methyl_fmr1.tolist()).
X = pd.DataFrame(df[['frac_hap_methyl_xci', 'fmr1_mean_repeat_length']]).
X_const = sm.add_constant(X) # Adds intercept term
model_sm = sm.OLS(y, X_const).fit().

In order to test association of ASD status to CGG repeat size and haplotype methylation fraction at *FMR1* we use a logistic regression model:

glm(Phenotype ~mean_repeat_length + frac_hap_methyl_fmr1, data = df, family = binomial).

### Possible mechanisms of complex SV generation

Junction sequences in the deletion and duplication breakpoints In Figure 3 can help propose possible mechanisms for the creation of the SVs. The sequence details of the breakpoints and their alignment to the reference are summarized in the supplemental information. In total we found signatures of the microhomology-mediated end joining (MMEJ) in the deletion in Figure 3C and the duplication in Figure 3E. We also found signatures of Non-homologous end joining (NHEJ) with short templated insertion in the duplication in Figure 3C. The other breakpoints did not show a clear mechanism explaining the junction sequences. The details of our findings for each subfigure is as follows.

In Figure 3A we only can detect the breakpoint locations of the inversion accurately. The duplication breakpoints are intersecting with segmental duplication regions which makes read mapping and exact location of the breakpoints inaccurate. The inversion in this example is very clean without an inserted sequence or presence of short or long stretches of homologous sequences. Therefore, mechanisms such as microhomology-mediated end joining (MMEJ) or non-allelic homologous recombination (NAHR) are unlikely to be responsible for this inversion.

The junction sequence of the duplication in Figure 3C contains an extra 7 base pair sequence between the upstream and downstream of the junction which is identically repeated upstream of the right breakpoint as well. This is very characteristic of **Non-homologous end joining (NHEJ)/alternative end joining** with short templated insertion; or a **replication-based mechanism** like FoSTeS/MMBIR where the polymerase transiently switches template nearby, copying a short stretch twice. For classic microhomology mediated mechanisms (MMEJ/MMBIR), we expect a short stretch (2–20 bp) that is shared between the downstream of left and upstream of right breakpoints which we lack in this case. Furthermore the NAHR (non-allelic homologous recombination) we expect a long stretch (>10–20 bp) homology between upstream of left and downstream of right breakpoints which we also lack. Therefore, MMEJ or NAHR is unlikely for these two breakpoints.

The deletion breakpoint however in Figure 3C is consistent with the Microhomology-Mediated End Joining (MMEJ). There is a 2 base pair sequence (GC) upstream of the left and downstream of the right breakpoints, one of which is deleted in the junction sequence together with the reference sequence in between. This is exactly what one expects from an MMJE mechanism.

We observe a 4 base pair sequence (ATTT) microhomology near the downstream of the left and upstream of the right breakpoints of the duplication in Figure 3E. We also observe a 4 base pair sequence (CCCC) in the junction between the two breakpoints. This sequence exists upstream of the right breakpoint, and could be a tiny insertion copied from nearby, but at 4 base pairs, it's too short to be definite. This combination of a few base pair microhomology and a small template insertion is the classic signatures of MMEJ or a related replication-based mechanism such as FoSTeS/MMBIR which also often leaves short microhomologies and small insertions at SV junctions.

For the deletion junction in Figure 3E however, we do not observe short repeated sequences at the breakpoint junctions (indicating the MMEJ) mechanism or long homologous sequences near the breakpoints (indicating NAHR mechanism).

### RNA extraction from whole blood samples

RNA was extracted from 500 μL or 250 μL of whole blood from human subjects (REACH000236 and subjects listed in Table S21). RNA was extracted using Zymo Direct-zol RNA Miniprep Plus kit (Zymo #R2073). Whole blood was diluted with 750 μL/375 μL DNA/RNA shield solution (Zymo #R1200-25). 12.5/6.25 μL of 20 μg/mL proteinase K (Zymo #D3001-2-5) and the entire mix was rotated at room temperature for 30 min. Afterward, TRI Reagent was added at 3:1 ratio and then loaded onto a Zymo-Spin IIICG column and processed according to manufacturer instructions (Zymo #R2073).

### Reverse transcription of RNA

RNA was converted to cDNA using the iScript cDNA synthesis kit (Bio-Rad # 1708891). 100ng–500ng of RNA was combined with 4 μL of iScript Reaction Mix and 1 μL iScript Reverse Transcriptase and water. cDNA synthesis was conducted on a thermocycler run at 5 min 25°C, 20 min 46°C, 1 min 95°C, and hold at 4°C.

## PCR and gel electrophoresis of *STK33* mRNA

cDNA template was amplified using various primer pairs (Figure S6; Table S22) for *STK33* or *GAPDH*. We used the Platinum SuperFi II PCR kit (Thermofisher # 12361010) using 5 μL superFi II buffer, 0.5 μL 10mM dNTPs, 0.5 μL Platinum SuperFi II DNA polymerase, 17.25 μL water, 0.5 μL of cDNA template, and 0.625 μL of 20 μM forward and reverse primers. Mixes were placed in thermocyclers at 98°C 30 s, and cycled for 35 rounds or 60 rounds at 98°C 10 s, 60°C 10 s, 72°C 1 min. A final extension was done at 72°C for 5 min followed by 4°C hold.

The 25 μL of PCR samples were mixed with 5 μL of 6X loading dye (Thermofisher #R0611) and loaded onto a 3% TAE agarose gel stained with GelGreen (Biotium #41005). For DNA ladder we used 100bp ladder (Thermofisher #SM0241). Gels were imaged with FluorChem E (Bio-Techne).

## PCR and iSeq 100 for *FMR1* mRNA

The *FMR1* mRNA sequence containing an allele-specific SNP (hg38 chrX:147,928,802 G/A) was amplified using PCR. We used the Platinum SuperFi II PCR kit (Thermofisher # 12361010) using 4 μL superFi II buffer, 0.4 μL 10mM dNTPs, 0.4 μL Platinum SuperFi II DNA polymerase, 11.2 μL water, 2 μL of cDNA template, and 1 μL of 10 μM forward and reverse primers (Table S21). The forward and reverse primers also appended Illumina i5 and i7 adaptor sequences and sample indices needed for downstream next generation sequencing of amplicons (Figure S12; Table S21). For every PCR reaction, the forward primer consisted of a 10 μM equimolar mixture of i5:1 through i5:6, while the reverse primer contained a sample specific index. Mixes were placed in thermocyclers at 98°C 30 s, and cycled for 45 rounds or 50 rounds at 98°C 10 s, 60°C 10 s, 72°C 1 min. A final extension was done at 72°C for 5 min followed by 4°C hold.

The 20 μL of PCR samples were mixed with 4 μL of 6X loading dye (Thermofisher #R0611) and loaded onto a 3% TAE agarose gel stained with GelGreen (Biotium #41005). For DNA ladder we used 100bp ladder (Thermofisher #SM0241). Gel bands at the expected amplicon size (246–251 bp), were excised with a scalpel and extracted using the QIAquick Gel Extraction Kit (Qiagen # 28706), and then further cleaned using AMPure XP Beads (Beckman Coulter #A63880).

Concentrations of cleaned amplicons were measured using the Qubit dsDNA BR assay (ThermoFisher #Q32850), diluted to 50pM in EB buffer, and loaded onto iSeq 100 Sequencing System (Illumina). Single-read sequencing was run at 110 cycles. The sample index was run at 8 cycles.

## Data processing of Iseq 100

Sample fastq files were aligned with HISAT2[68] to a custom genome containing two reference contigs matching the PCR amplicon region (Table S23). The sequence of the two reference contigs only differed by the allele-specific SNP (hg38 chrX:147,928,802 G/A). The allele-specific *FMR1* expression was calculated by the ratio of number uniquely aligned reads that mapped to the A allele versus the G allele (Table S13).

## Analysis of breakpoint sequences of DUP-DEL events

The following results describe the breakpoint sequences of the DUP and the DEL events described in Figure 3 for the purpose of inferring the underlying mutational mechanism. The sequences provided include the left and right boundary sequences of the grch38 reference genome with a pipe (|) designating the breakpoint position. The "junction" sequence of the DEL and the DUP is the actual breakpoint sequence obtained from long reads spanning the junction. Microhomology sequences are underlined and short insertions of sequence at the junction are highlighted with *italic font*.

For classic microhomology mediated mechanisms (MMEJ/MMBIR), we expect a short stretch (2–20 bp) of sequence that is shared between the left and right breakpoints. For Non-allelic homologous recombination (NAHR) we expect a longer stretch (>10–20 bp) of homology.

Breakpoints in Figure 3A.

Duplication breakpoints: In this example, the boundaries of the inverted duplication are located in dense clusters of segmental duplications and likely involve a rearrangement by non-allelic homologous recombination (NAHR).

Deletion breakpoints: The deletion breakpoint is clean without an inserted sequence or presence of short or long stretches of homologous sequences. Therefore, the deletion likely occurred by a distinct repair mechanism such as non-homologous end joining (NHEJ).

Deletion left boundary:
GCTGCAGTCTTCATTAGTTAACCTTAAACCTTTACCTCAAAGAAAGGTATCACTTGAAGACCAACTGTATTAGACTGTTTTCATGC TGCTGATAGACAT | AACCAAAGCTAGGAACAAAAAGTGGTTTAAGGGCGGGAGCAGTGGTTCATGCCTGTAATCTCAGCACTTTG GAAGGCCAAGGTGGGCGGATCACAAGGTCA.

Deletion right boundary:
CTCCCAAAGTACTGGGATTACAGGCGTGAGCCACTGTTCCCGGCCCAGCAAGTTTTTTCATGTCTGTACTTAGAAGGGCACTAA TCTTATCATGAGGTT | CCCACCCTTATGACCTCATCCAAACCATATTACCTCACAAAGACCCTGTCTCCAAATGCTATCATATTGGG GGTTGGGGCTTCAACATAAATTTTAGGGGA.

Deletion junction:
reverse complement of ref right breakpoint on the right side | left breakpoint on the right side.

CCCCTAAAATTTATGTTGAAGCCCCAACCCCCAATATGATAGCATTTGGAGACAGGGTCTTTGTGAGGTAATATGGTTTGGATGA
GGTCATAAGGGTGGG | AACCAAAGCTAGGAACAAAAAGTGGTTTAAGGGCGGGAGCAGTGGTTCATGCCTGTAATCTCAGCACTT
TGGAAGGCCAAGGTGGGCGGATCACAAGGTC.

Conclusion: no microhomologies were found near the deletion breakpoint.

Breakpoints in Figure 3C.

<u>Duplication breakpoints:</u>

The sequences around the duplication breakpoints in Figure 3C are provided below. The sequences before and after the break-points are separated with a pipe and if an extra sequence exists for the sample genome it's given in between two pipes. An alignment of the sample junction sequence to the left and right breakpoint reference sequences is also provided below.

Duplication left boundary:

TTTGTATTTTTGTAAGAGACGGGGTTTCACCATGTTGGCCAGGCTGGTCTTGAACTCCTGACCTCAGGTGATCCGACAGCCTTG
GCCTCTGAAAATGCT | AGGATTACAGGTGAGAGCCACCACACCCAGGTAATTATTATTTTTTTGAGACCGGGTCTCACTCTATTACC
CAGGATGGAGTGTAGTGTTGTGATCATGGCT.

Duplication right boundary:

GTGGTTATGTTACTAGCAACAATAAGCCGCTACTTTTGTTGAAACAATAAAACTGCATTTTATTTCTGAAATACAAACTATTACTGA
TT<u>CTTACAC</u>AAT | GTCAAAAATGTTCTAGGCTATTCTGCTTTTGTTTAGACTATCAAGAGATACTCTAGGCTATAATTCAC<u>TTTTTTTTT</u>
CTCCTACAGATTCTGAAGATTGCT.

Duplication junction:

AGTGGTTATGTTACTAGCAACAATAAGCCGCTACTTTTGTTGAAACAATAAAACTGCATTTTATTTCTGAAATACAAACTATTACTG
ATT<u>CTTACAC</u>AAT | *CTTACAC* | AGGATTACAGGTGAGAGCCACCACACCCAGGTAATTATTATTTTTTTGAGACCGGGTCTCACTCT
ATTACCCAGGATGGAGTGTAGTGTTGTGATCATGGC.

Conclusion: no microhomologies were found near the deletion breakpoint. We do find that a 7 bp sequence of the left breakpoint has become duplicated by the rearrangement. Short templated insertions are a characteristic of **Non-homologous end joining (NHEJ)/alternative end joining** or a **replication-based mechanism** like FoSTeS/MMBIR where the polymerase transiently switches template nearby, copying a short stretch twice.

<u>Deletion Breakpoints:</u>

Deletion left boundary:

AGTAGCAGCTGGGATTACAGGAGCACGCCAGCGCCACCATGCCCAGCTAATTTTTGTATTTTTTTTAGTAGAGAAGGGGTTTCA
CCATGTTGGTCAGG<u>GC</u> | TGGTCTTGAACTCCTGACCTCGTGATCCGCCCGCCTCAGCCTCCCAAAGTATTGGGATTACAGGCATGA
GCCACTGCGCCCGGCCTACCGGCCTAGTATTC.

Deletion right boundary:

CTGCAGGCATGTGCCTCCATAGCTGGCTAATTTTTGTATTTTTTGTAGAGACAGGATCTCACTGTGTTGCCCAGGCTGGTCTTGA
ACTCCTGACCTCAA | <u>GC</u>CATCCTCGTGCCTCAGCCTCCCAAAGTGCTGGAATTACAGGCATGAGCCATTGCGCCTGTCATTGTCT
TTTAAATTACAGCAATTCCAATGACAGCAAA.

Deletion junction:

AAGTAGCAGCTGGGATTACAGGAGCACGCCAGCGCCACCATGCCCAGCTAATTTTTGTATTTTTTTTAGTAGAGAAGGGGTTTC
ACCATGTTGGTCAGG<u>GC</u> | CATCCTCGTGCCTCAGCCTCCCAAAGTGCTGGAATTACAGGCATGAGCCATTGCGCCTGTCATTGTCT
TTTAAATTACAGCA<u>TT</u>CCAATGACAGCAA.

Conclusion: The left and right breakpoints share a GC dinucleotide sequence consistent with Microhomology-Mediated End Joining (MMEJ).

Breakpoints in Figure 3E.

<u>Duplication breakpoints:</u>

Duplication left boundary:

TACATGTTGAACACATTACATGTTGAACACATTTATTTACATGTTGAACACCTTACAACTGCAGTCTTCCACTAATTCCTAACTTTT
GATTAAACCT | <u>ATTT</u>CTCTTTTTCTTGGCTATAGAGAATTACTGGTTTGGAGGGCAACTCATGCCATTTAAGTTCCACTGTACTCTAA
GTCTCACACAACTCCTTCTCATAATC.

Duplication right boundary:

GCCCAGAACAAGACCCCATCTCTTAAAAAAAAAAGAAAGTAAAAAATAGTTACTAGGTGAGTGGAAGAGTGAATATACTAGGGTAA
TACAGTAC<u>ATTTG</u> | TCAGACAAGGGGGACTTGAGGTTGGTAATCATGA<u>ATTT</u>AAAGTAGACCAGTAAGAATAGTCAGCTATTTTTTTC
TAGTCAT<u>TTT</u>GGAACTGTACAGGTAGAGAA.

Duplication junction:

CTGCCCAGAACAAGACCCCATCTCTTAAAAAAAAAAAGAAAACTAAAAAAATAGTTACTAGGTGAGTGGAAGAGTGAATATACTA
GGGTAATACAGTAC<u>ATTTG</u> | *CCCC* | <u>ATTT</u>CTCTTTTTCTTAGCTATAGAGAATTACTGGTTTGGAGGGCAACTCATGCCATTTAAGT
TCCACTGTACTCTAAGTCTCACACAACTCCTTCTCATA.

Conclusion: We observe a 4 bp sequence (ATTT) microhomology near downstream of the left and upstream of the right breakpoints. We also observe a 4 bp sequence (CCCC) in the junction sequence between the two breakpoints. These are signatures of MMEJ or a related replication-based mechanism such as FoSTeS/MMBIR which also often leaves short insertions at SV junctions.

<u>Deletion</u> breakpoints:

Deletion left boundary:

CATGTACTACTCAATTAATATGAGATTAAATCACTCACCTTATCAAGTTCACTCGTCAGCTTTTTATTTTCTTCAGTTAACAACACTT
TTTCTCGTTCA | TATTGTTGTTTGAACTCACTTTCAAATTCCTCCCTTTCACTTTGACTAACACAATTCAAAACACAAAAGGAAAAAGG
GGAATTAAAGGTCATTTGAAAACT.

Deletion right boundary:

AACTTCCAGTCCTGTAAGCTTCATTCATGGTAAGTGCCTTATATAGGTGTACCCTTTTTAAAAAAAATCTTGTATATAATCATATAA
CAATACCTTTCT | CAGAAAATATCCCTGGTGTTAAGCAATGCATGACTGTAATTCCCTGCAATAGTTTTACTCAGAATAAGAAACTAT
GCATCCACTAAATATGACTTACAAGG.

Deletion junction:

GCATGTACTACTCAATTAATATGATTAAATCACTCATATCAAGTTCACTCGTCAGCTTTTTATTTTCTTCAGTTAACAACACTTTTCG
TTCA | CGAGAAAAATTACATGACTGTAATTCCCTGCAATAGTTTTACTCAGAATAAGAAACTATGCATCCACTAAATATGACTTACAA
GG.

Conclusion: For the deletion junction we do not observe short homologies at the breakpoint junctions consistent with NHEJ.

## QUANTIFICATION AND STATISTICAL ANALYSIS

The T-tests performed in Figure 2B, are two-sided with $N$ = 55. The details of the association models and skewness calculations in Figure 5 are presented in the STAR Methods section. The statistical models and the covariates used for burden test analysis and the R-squared calculations in Figure 6 are presented in the STAR Methods section as well as the results section.

