## [Data S1. Transparent peer review records for Mortazavi et al. · Cell Genomics]

Summary

Initial submission: Received : July 17, 2025

Scientific editor: Laura Zahn and Haoyang Wei

First round of review: Number of reviewers: 3
Revision invited : December 29, 2025
Revision received : January 2, 2026

Second round of review: Number of reviewers: 2
Accepted : February 12 2026

Data freely available: Yes

Code freely available: Yes

This transparent peer review record is not systematically proofread, type-set, or edited. Special characters, formatting, and equations may fail to render properly. Standard procedural text within the editor's letters has been deleted for the sake of brevity, but all official correspondence specific to the manuscript has been preserved.

Referees' reports, first round of review

Reviewer #1: In this study, Mortazavi and colleagues utilized long-read whole genome sequencing (LR-WGS) on an autism spectrum disorder (ASD) cohort and demonstrated that it significantly enhances the detection of structural variants (SVs) and tandem repeats (TRs), uncovering novel de novo mutations and a recurrent class of complex rearrangements. They performed a burden analysis showing that rare variants explain a modest but meaningful portion of ASD heritability. Overall, this work established LR-WGS as a powerful tool for resolving complex genetic variation. The study is significant and valuable, the manuscript is well written, and the data analyses are rigorous. However, there are some comments below that need to be addressed to enhance the manuscript.

1. Why were two different long-read platforms used? What are the known differences between them that could impact the results?

2. The authors must include any analysis performed to ensure that there were no platform-specific batch effects.

3. Were there any significant differences in the number of SVs and TRs detected by each platform?

4. Please include a supplementary figure showing the distribution of coverage for each platform to confirm that differences in variant counts are not simply a result of different sequencing depths.

5. What was the rule for merging call sets? If an SV was called by both long-read and short-read, which breakpoints were used?

6. What was the rationale to prioritize TR calls over SV calls?

7. Provide individual-level statistics for the variants. What was the average number of SVs per individual? Is there a significant difference in the number of SVs per individuals between cases and controls? Is there any correlation between the number of SVs per individual and other variables such as sex and age?

8. Cite the previously published SV call set from short-read (SR)-WGS under the section "Comparison of SVs and TRs between LR-WGS and SR-WGS".

9. It would be helpful to define the criteria for "constrained genes" in the Results section. How many constrained genes were used for all downstream analysis? Provide a supplementary list.

10. Define "novel" dnSVs. Is it in terms of a new ASD gene, a new class of SVs, or the specific variant itself is novel?

11. The authors used a filtering threshold of at least 3 phased long reads ($AD \geq 3$) supporting the variant in the proband, but only at least 1 phased long read ($AD \geq 1$) supporting the reference allele in each parent. Why was there a difference in the read depth thresholds used?

12. Clarify if the 3 dnSVs were identified in cases or controls.

13. Expand on the function of SH3PXD2A. What pathway is it involved in, and how does it relate to ASD pathogenesis.

14. Axes labels for Figure 2A are missing.

15. How many DUP-DEL-like rearrangements were identified in the entire cohort? Are the three cases presented here the only examples, or are they the most compelling examples from a larger set?

16. For each of the three DUP-DEL cases described, what is the clinical status of the individual? It is a crucial piece of information to link these complex rearrangements to specific phenotypes.

17. Given the non-significant or modest p-values for many of the burden analyses, what is the statistical power of the study to detect a real effect? What sample size would be required to achieve a $p < 0.05$ for the burden of SVs in constrained genes, assuming the observed effect size is real? Include a power analysis.

18. This manuscript is a valuable contribution to the field of genomics, demonstrating the power of LR-WGS to uncover complex genetic variation. However, the overall tone feels more like a showcase of a new technology than a focused study on the genetic architecture of ASD. The narrative could be strengthened by a deeper focus on the condition. Especially because the authors have the word "autism" in the title. Some points to incorporate are listed below:

a. A more detailed clinical and demographic characterization would greatly strengthen the paper's connection to ASD. What were the precise inclusion and exclusion criteria for both the ASD cases and the unaffected controls? What are the key clinical phenotypes most frequently observed in the ASD case group?

b. The analysis of FMR1 is a technically impressive demonstration of LR-WGS capabilities, but its relevance to the primary topic of ASD is only addressed briefly at the end of the paragraph with a non-significant finding. This makes the section feel somewhat disconnected from the central narrative.

c. The author uses a constrained gene list for downstream analysis, but does not explicitly incorporate the widely-used SFARI gene list. A burden test on SFARI genes would provide a more targeted, disease-specific analysis. What % of the constrained genes are ASD genes?

Reviewer #2: The manuscript by Milad Mortazavi and colleagues describes the use of long-read sequencing to characterize genetic variation in autism spectrum disorder (ASD). The genomes of 267 individuals from 63 families have been sequenced using either PacBio or Oxford Nanopore sequencing. Variant call sets are compared to previously generated short-read results. Specific complex SVs are described in greater detail. Methylation analysis is performed focusing on FMR1, showing that gray-zone alleles exhibit increased methylation. Overall, the conclusion of the paper is that rare variants, and especially SVs, explain 6.2% of ASD heritability in this cohort.

This is a well defined cohort and long-read sequencing of such cohorts are of general interest, making the general theme of the paper interesting and timely. However, there are also a number of issues with this study that will need to be addressed in order to publish the manuscript in a high impact journal.

1) The introduction does not really bring up any of the studies that have been done previously to show the benefits of long-read sequencing in comparison to short-read sequencing, with regard the improved SV calling. Since it is highly relevant to the paper, it should be added, with proper references to the most prominent studies.

2) The sequence coverage is very low for most of these individuals, and the sequence data is generated by "older" generations of PacBio (Sequel IIE) and Oxford Nanopore (R9) flow cells. Especially the low coverage is really the main issue with this study. While the ability to detect SVs is still decent with limited coverage of long reads, it is too low here for many individuals to really call variants reliably i large parts of the genome. Low coverage will also reflect poorly in large CNV regions where breakpoints are in long repeats, i.e. where detection is based on fluctuations in read counts/coverage. This also explains why so many short-read only SVs were detected. Compared to other long-read benchmark studies the number of SVs identified is very low, as a result of low sequence coverage and possibly in combinations with calling parameters. Overall, it is questionable whether this level of coverage will give very useful new insights beyond the off finding of some specific individuals with interesting rearrangements or methylation profiles.

The authors discuss long-read sequencing as if it is one method, while they have in fact used two different methods for long-read sequencing. These methods will find different sets of SVs and SNPs, have slightly different strengths and weaknesses in estimating VNTR length, different algorithms for 5mC calling etc.

Since read-length may vary greatly for long-read methods depending on how the DNA was treated and how libraries were constructed, it should be stated in the text what the average read-length and distribution are for each of the long-read platforms. Now these quite informative numbers are only found in Supplementary Table 1. The mean read length for some of the nanopore is actually quite low and will affect the ability to call variants.

The first paragraph in the results section is slightly confusing in terms of what individuals/families were sequenced. It seems to be a mix if incomplete families, trios and quads (and possibly larger families?), but this is impossible to decipher from the way it is currently written. It is clear from Table S1, but should be written more clearly also in the text.

The end of the first paragraph and start of second paragraph in the section about de novo SVs are unclear in terms of number of de novo events identified and confirmed. 65 candidate dnSVs were identified and SR-WGS was used for validation and genotyping. But in the next sentence it says 6 were unique to SR-WGS - clearly not referring back to the previous sentence. Some clarification is required for the reader to actually follow the reasoning here.

The calling and validation of dnSVs is also a bit unclear in the methods section. It is stated that high confidence de novo SVs requires allele depth >3 in the offspring, and zero AD in both parents, each of whom must have 1 or more phased long read supporting the reference allele in both haplotypes. With this definition it is unclear how a de novo duplication can be called? Clearly a duplication allele can not have 0 read depth in both parents as one of the alleles must be the one that is duplicated?

It also seems strict to require phasing in the long-read data since phasing often falls apart in rearrangement regions, especially if surrounded by long repeats, which is often the case for SV regions. The definition that both parents must have one or more phased long read supporting the reference allele on both haplotypes also seems like it would exclude certain events, e.g. where a parent carrying an inversion haplotype has an off-spring with de novo del or dup in that region (a mechanism well established in e.g. the Williams region).

Were there no LR-WGS dnSVs called in regions where SR-WGS could not be used for validation due to issues with mapping? What happened to such candidate dnSVs in this study?

It is unclear for the rearrangements described in Figure 3 what evidence there is for the rearrangement architecture. Are there breakpoint spanning reads that confirm the order and orientation of sequences? As an example (but valid for all complex SVs in figure 3), in Figure 3b, is there evidence that the deletion within the tandem dup is upstream of the complete copy? Are there breakpoint reads that can be used to exclude that one copy is inverted? Is there any evidence that this is a TAN-DUP-DEL rather than two smaller duplication events (i.e. instead of initial tandem dup followed by del, there would be two smaller duplications events (and no deletion))? Are there details around the breakpoints that may help elucidate the mechanism (e.g. inserted bases, microhomology etc)?

For the FMR1 analysis, it is unclear how many females that actually pass the requirement of having at least 3 phased reads per haplotype, and how many end up in each of the three groups (gray-zone, short, intermediate). It would be helpful if the numbers are stated in the text.

Reviewer #3: Mortazavi et al. present a study in which they sequenced 243 individuals from 67 families using long-read sequencing technologies. The authors perform multiple analyses, including calling structural variants (SVs) and tandem repeats (TRs), comparing long-read derived SVs with previously generated short-read SV calls, detecting potentially relevant tandem variants, identifying SV signatures, assessing methylation near FMR1, and evaluating the burden of variants in autism spectrum disorder (ASD). The manuscript leverages a well-established cohort and applies advanced sequencing technologies; however, there are several important limitations and methodological concerns that warrant discussion.

Strengths

This study benefits from a uniquely valuable cohort: families with ASD that have been extensively characterized over many years in the Sebat Lab. Applying long-read sequencing to such a resourceful dataset represents a meaningful step toward discovery.

The integration of diverse analyses spanning SV calling, TR detection, methylation analysis, and burden testing provides a broad perspective on the potential contributions of structural variants and tandem repeat variants to ASD.

The paper addresses a timely and compelling topic.

Major Concerns

1. Sequencing Depth and Platform Heterogeneity

The most significant limitation is the shallow sequencing depth. Reported average coverage (Table S1) is only 3.8x for PacBio and 10.7x for ONT. Such low coverage substantially reduces variant detection sensitivity and raises concerns about both false negatives and false positives. Figures S1 and S2 further highlight the suboptimal performance associated with these specific depth levels. While the authors attempt to adjust for coverage by including depth as a covariate, such modeling cannot fully substitute for adequate raw coverage. Compounding this issue, different samples were sequenced on two distinct platforms with very different error profiles and read characteristics. Mixing PacBio and ONT data introduces additional variability, making it difficult to distinguish biological signal from technical noise. For example, a variant absent from a PacBio sample may be missed due to very low coverage, while ONT-specific artifacts could lead to spurious calls. Yet, the study does not include a direct within-sample comparison of PacBio versus ONT even for the one sample described as "high coverage" (40x ONT, 15x PacBio), which still falls short of standard high coverage for SV analysis. I would expect at least 30x coverage

for PacBio to be a standard coverage with high coverage being substantially higher (e.g., 60x). Why was a direct PacBio-ONT comparison on the "high-coverage" sample not included? How might platform-specific error modes (e.g., ONT homopolymer-associated errors) bias detection of certain SV classes? Are the covariate models robust enough to adjust for both within-platform and cross-platform biases?

2. Choice of Variant Calling Tools

The study appears not to have used the most widely adopted long-read SV/CNV callers. For PacBio HiFi data, callers such as pbsv and hificnv are standard. Other popular tools such as cuteSV and SVIM were also not considered in the analyses. Why were PacBio-specific tools (e.g., pbsv, hificnv) not used? Was this a deliberate methodological choice? Were cuteSV or SVIM considered? Were existing benchmarking studies (e.g., GIAB truth sets) used to guide tool selection? Could omission of these callers disproportionately affect detection of certain SV classes (e.g., tandem repeats, insertions, large deletions)?

3. Validation of De Novo Variants

The criteria for defining de novo variants appear underdeveloped. With PacBio coverage averaging only 3.8x, a threshold of allele depth >3 risks missing genuine de novo events. Moreover, it is not clear what orthogonal validation strategies were utilized (e.g., Sanger sequencing, SNP arrays, or independent sequencing technologies). What additional methods were used to confirm candidate de novo variants? What fraction of potential de novo events might be missed under the current allele depth and coverage parameters?

4. Burden Model

The description of the burden model raises questions. If single-nucleotide variants (SNVs), polygenic risk scores (PRS), and principal components (PCs) are derived from ONT data, this is problematic, as ONT is not well suited for high-quality SNV detection. Clarification is needed as to whether these inputs were instead derived from orthogonal datasets.

Additional Points for Clarification

Figure 1A, 1B, 1C: Please report not only absolute variant counts but also the proportion validated by orthogonal methods, as discovery rates differ by platform, but error rates may as well.

Page 5 statement ("Structural modeling using AlphaFold 23 suggests a substantial alteration to the C-terminal structure of the mutant protein"): Please clarify the specific structural differences and their biological relevance. Likewise, in Figure 2D, clarify what difference is being illustrated. Is there a statistical model you can apply here?

Figure 2A, 2B: What method was used to estimate copy number?

Paper Title ("diverse functional consequences"): Please specify what these consequences are as currently, the title claim is too vague.

Methylation analysis of FMR1: This section seems disconnected from the main analyses. Consider moving it later in the manuscript, expanding it to include a broader methylation analysis (e.g., imprinting), or clarifying its role in the overall narrative.

Summary

This manuscript leverages a uniquely valuable ASD cohort and applies long-read sequencing in a manner that has clear potential for discovery. However, limitations in sequencing depth, lack of direct cross-platform benchmarking, omission of platform-specific variant callers, and insufficient validation strategies significantly weaken the strength of the conclusions. Addressing these issues, particularly by performing direct PacBio-ONT comparisons, incorporating optimized SV callers, and validating key findings, would substantially increase the robustness and impact of this work.

Authors' response to the first round of review

Reviewers provided many comments that were helpful in addressing specific weaknesses in the paper. We were pleased to submit a revised manuscript that addresses the reviewers key

concerns.

To summarize, our study demonstrates that LR-WGS substantially improves the detection of gene-disrupting structural variants and tandem repeats, particularly those at smaller scales (<1,000 bp) that are often missed by SR-based methods. LR-WGS also offers distinct advantages with respect to determining the functional consequences of SVs. Long reads provide precise resolution of fine-scale structural features and complex rearrangements. Phase information facilitates detection of somatic mosaicism. Furthermore, joint analysis of phased genetic variants and DNA methylation enables functional characterization of variants in *FMR1* and imprinted genes. Collectively, rare variants explain ~7% of heritability in this sample, but application of LR-WGS to larger cohorts is needed to have reasonable power to detect genetic associations and refine these estimates. To capture the essence, the manuscript has a new title: “Long-Read Genome Sequencing Improves Detection and Functional Interpretation of Structural and Repeat Variants in Autism”

A theme throughout this paper is: When one makes the leap from short reads to long reads, one experiences a dramatic shift toward having a deep dataset of structural and tandem repeat variation that includes sequence level breakpoints and structural information, phasing and methylation information produced from a single assay. When one applies LR-WGS at scale to a disease, what is immediately apparent as “game changing”, is not the discovery of a new gene (finding new autism genes is commonplace in modern genetics). The most novel aspect is precisely the variety of functional effects on genes that can be characterized.

The most substantial additions were made in response to concerns shared by multiple reviewers:

- At the suggestion of reviewers 1 and 3, we have expanded the joint analysis of SVs and DNA methylation by investigating rare variants in imprinted genes. A maternally-inherited deletion of a maternally-expressed gene *ADNP2* was found in one family, a pattern that is consistent with an imprinting disorder. To highlight more clearly methylation of the imprinting control region of *ADNP2* we have sequenced this trio more deeply with the new PacBio Revio instrument at UCSD. There is also genetic evidence from a recent study supporting *ADNP2* in neurodevelopmental disorders (PMID: 35982160). However, further studies are needed to determine what, if any, phenotypes are associated with *ADNP2* loss of function in humans.

- Much greater detail is provided on the breakpoint sequences and structures of complex DUP-DEL SVs.

- Sequencing platforms and coverage (not to mention sample size) are well-known issues inherent to large-scale studies of rare variants. Our study incorporates standard, established approaches to mitigate them. We have revised the manuscript to more explicitly describe our study design and the safeguards we employed to address these factors. Considerable attention was given to this subject in the reviews and in our response to reviewers

manuscript revision. While these concerns are primarily relevant to the last figure (Fig. 6), these technical issues are broadly relevant to researchers that are looking to apply LR-WGS in genetic association studies. Thus, it will be of interest that we have performed comparisons of platforms with respect to detection of SVs and TRs and provided detailed rationale for the statistical methods used to control for these factors. We provide detailed responses to each comment below.

Reviewer #1 : In this study, Mortazavi and colleagues utilized long-read whole genome sequencing (LR-WGS) on an autism spectrum disorder (ASD) cohort and demonstrated that it significantly enhances the detection of structural variants (SVs) and tandem repeats (TRs),

uncovering novel de novo mutations and a recurrent class of complex rearrangements. They performed a burden analysis showing that rare variants explain a modest but meaningful portion of ASD heritability. Overall, this work established LR-WGS as a powerful tool for resolving complex genetic variation. The study is significant and valuable, the manuscript is well written, and the data analyses are rigorous. However, there are some comments below that need to be addressed to enhance the manuscript.

1. Why were two different long-read platforms used?

This project began in 2017 as an NIH grant to expand the genetic variation captured in our autism cohort by low-coverage WGS with the oxford nanopore (ONT) platform. The proposal also planned to evaluate new technologies as they became available. Thus, the evolution of this project has paralleled the evolution of sequencing technologies, which have developed rapidly during this time. In 2019, PacBio released the Sequel II instrument which made circular consensus sequencing (HiFi) widely accessible. Based on our pilot studies (see Figure S2 below), the higher base calling accuracy of HiFi sequencing provided superior SV calling accuracy for a given level of coverage. For the 2nd half of the project, we purchased a Sequel II instrument.

What are the known differences between them that could impact the results?

There are 3 major differences. (1) Nanopore technology has, historically, had a lower opportunity cost, i.e. the cost of the GridION instrument and flow cells were significantly more affordable than PacBio instruments available at the beginning of the project. (2) ONT is capable of producing longer reads than PacBio

<https://nanoporetech.com/blog/news-blog-kilobases-whales-short-history-ultra-long-reads-and-hi-gh-throughput-genome> . (3) Lastly, at the beginning of the project, both platforms had similar base calling accuracy (10% error rate), but the release of the HiFi technology in 2019 brought the base calling error rates of PacBio to <1%. Below we elaborate on these differences based on our experience.

Read lengths : In this study, when manufacturer recommended sample prep protocols were applied to existing collections of frozen DNA, read lengths obtained with PacBio (11,300 bp on average) were in fact longer than read lengths in our ONT data (5,600 bp)(see Table S1, Figure S1). There are a variety of technical factors that explain why ONT read lengths were not as long as advertised. (1) the “whales” that ONT promotional materials are referring to are seen largely with cell-line derived high molecular weight DNA, which is not representative of existing collections of frozen patient DNA samples. (2) The size selection step in the original ONT manufacturers protocol was likely not the most efficient method possible. Lastly, (3) ONT sequencing of very long DNA fragments in a solution is less efficient than sequencing of smaller fragments because long DNA has a high tendency to form knots and clog nanopores <https://pubmed.ncbi.nlm.nih.gov/30960068/> . Getting significant coverage of ultralong reads on a sample is a lot of work to achieve. On existing collections of frozen DNA, it's nearly impossible.

This is one of the reasons we switched to PacBio in 2019.

Figure S1. Read length and coverage distribution of subjects in the cohort. (A) Read length distribution stratified by platform.

SV calling accuracy is greater for PacBio for a given level of coverage. Using a truth set created on high-coverage data from one sample (REACH000236), we evaluated accuracy of SV calling of both platforms as area under the sensitivity-FDR curve for varying levels of coverage. Coverage required to achieve reasonable accuracy (AUC ~ 0.8, orange curves in Fig. S2A and S2B) was 5X and 10X for PacBio and ONT respectively. Thus even if the coverage of PacBio was lower than the ONT data by a factor of 2, similar accuracy could be achieved. 5X and 10X was the target level of coverage for PacBio and ONT. Mean coverage of the final PacBio and ONT datasets that was achieved was ~4X and ~10X respectively.

Figure S2. Benchmarking snoopSV (an in-house genotyping method) to detect allele depth (AD) and assign quality metrics to SVs.

(A) Sensitivity vs. FDR for HiFi data. (B) Sensitivity vs. FDR for ONT data. Area Under the Curve (AUC) for each curve is annotated in the figures. FDR: False Discovery Rate.

This explains how the technologies differ, how they have evolved, and the basis behind the choice of platforms and coverage. We address additional points in response to reviewer 3 comment 1C.

2. The authors must include any analysis performed to ensure that there were no

platform-specific batch effects.

One can successfully address batch effects, but cannot deny their existence. As we have demonstrated in previous studies (see for example Figure S3 in PMID: 40791719), differences in SV detection by platform are ubiquitous, and they represent an important confounder. Platform effects behave similarly to population stratification. If a variant is detected by platform A and not by platform B, when the 2 platforms are combined, spurious genetic associations can arise, particularly if the case-control balance differs between the 2 platforms. While these are major confounders, they are readily addressed in the same way that population stratification is addressed: by (1) ensuring that platforms are analyzed separately and are not combined in a single regression model and (2) additional technical confounders such as coverage are controlled for in statistical models.

In Figure S4 (and in our response to the following comment #3), we show that SV detection differs by platform. We applied a standard approach used in GWAS to address heterogeneity of platforms and cohorts. We performed stratified meta-analysis, where the sample is separated based on the major strata (platform). Analysis is performed within platform and then the results are combined by meta-analysis. As described in our recent study of CNVs in the psychiatric genomics consortium (Shanta et al. 2025), when multiple platforms are combined in a single logistic regression model, platform specific CNVs give rise to spurious associations. A stratified meta-analysis eliminates the spurious signals (See Figure S3 in Shanta et al. 2025) attributable to platform.

Thus appropriate controls for platform, batch and coverage are incorporated into the analysis. We realize the importance of these details, and we have now provided more detailed descriptions of how we control for major confounders.

1) Meta-analysis stratified by platform

2) Additional batch effects are controlled for by sequencing family members in parallel, and association is tested within family by conditional logistic regression

3) clogit regression models further control for coverage and genome wide SV burden to account for within-platform and within family variation in SV detection between samples.

4) Inclusion of intronic and intergenic categories of functional consequence represent a control to capture a spurious burden signal. As we have demonstrated here and in previous studies (Brandler et al. 2018), the burden of intronic and intergenic SVs, as a whole, do not show association with ASD (Fig. 6).

5) Orthogonal validation of exonic SVs called by LR-WGS is obtained in the existing >30X SR-WGS dataset

3. Were there any significant differences in the number of SVs and TRs detected by each platform?

The number of SVs and TRs detected differs between platforms (Figure S4). Since coverage of the samples on PacBio HiFi (~4x) was on average less than the ONT platform (~10x) (Figure S1), the number of SVs and TRs (>50 bp in size) detected are different on the two platforms. Given that total SV burden is strongly predicted by coverage within platform, including coverage as a covariate in the analysis is effective for adjusting for this confounder. We have applied the standard approaches described above.

Figure S4. Number of structural variations found for each subject as a function of sequencing coverage and stratified by platform. (A,D) Number of non-TR SVs with sample quality greater than 20. (B,E) Number of TR regions with at least 50 bp deviation, at least two supporting reads and genotyping quality greater than 0.9. (C,F) Total number of SVs in non-TR and TR regions.

4. Please include a supplementary figure showing the distribution of coverage for each platform to confirm that differences in variant counts are not simply a result of different sequencing depths.

We have added Figure S1 illustrating the distributions of coverage, as well as read length metrics for each platform.

Figure S1. Read length and coverage distribution of subjects in the cohort. (A) Read length distribution stratified by platform. (B) Coverage distribution of subjects in the cohort stratified by platform.

PacBio HiFi has greater SV calling accuracy for a given level of coverage (Fig. S2), which is

attributable to the higher base calling accuracy and higher alignment qualities of the corresponding reads. Hence coverage is not the only factor that creates platform-related differences.

5. What was the rule for merging call sets? If an SV was called by both long-read and short-read, which breakpoints were used?

We first merge the Sniffles2 and LUMPY call sets. Call sets are merged for each SV type separately. For deletions, duplications and inversions which have two breakpoints per SV, we merge the SVs using bedtools intersect with 50% reciprocal overlap to detect the SVs which are called by both call sets. For the SVs which are present in both call sets we use the Sniffles2 breakpoints (due to higher accuracy). Since Lumpy does not detect insertions, we use Sniffles2 insertions for the long-read SV call set. To avoid over-filtering, we apply a more lenient criterion at this stage, requiring at least one supporting read across all subjects for an SV to be retained. Since tandem repeats (TRs) were genotyped separately, we exclude SVs overlapping TR regions, defined by UCSC Table Browser annotations, by more than 50% reciprocal overlap. The resulting long-read non-TR SV call set was then merged with a previously published short-read SV call set ²¹, keeping track of the platform of origin (long-read or short-read) for each SV. Similarly to what mentioned before, for deletions, duplications and inversions we merge the SVs using bedtools intersect with 50% reciprocal overlap to identify SVs which are detected by both platforms. For these SVs we use long-read breakpoints (due to higher breakpoint-level accuracy of long-read SV calls). For insertions, we use SV length to extend the insertion breakpoint in both directions symmetrically by half of the insertion length to define two pseudo breakpoints and similar to what was done for the other SV types, we use bedtools intersect with 50% reciprocal overlap to merge the SVs.

The scripts for merging within long read callers and for merging long read and short read call sets can be found in the analysis code github.

6. What was the rationale to prioritize TR calls over SV calls?

We assume that “prioritization” refers to our choice of genotyping methods (Sniffles vs LongTR). Apologies that this sentence may have been unclear. It case it was unclear, the has now been revised.

For SVs highly overlapping with TR regions (>50% reciprocal overlap), the SV call almost invariably represents an expansion or contraction of the TR. For these variants therefore, we rely on the TR genotyper LongTR ²², and TR SV calls (>50% reciprocal overlap) were excluded from the SV call set (non-TR SVs, Fig. 1). The resulting SV calls were then merged with existing short-read WGS SV call sets from a prior study with the same subjects ²³.

We have also further elaborated on this step of the data processing: A key requirement of our analysis workflow is that we have TR genotyping methods that work well on multiple platforms. A large fraction of the deletions and insertions >50 bp in length are located entirely within individual TR elements, and within TRs, there is much greater variability in the read alignments within and between platforms. When there is a >50 bp expansion/contraction of the TR, the aligner minimap2 may introduce multiple gaps in the alignment with significant variability in the alignment between reads. SV-calling algorithms that rely on breakpoint positions from sequence alignment thus have low accuracy for TR variants. By contrast, specialized TR genotyping approaches such as LongTR can estimate the total length of the repeat element without relying on the specific placement of gaps by minimap2. In contrast to sniffles calls, which have low concordance between HiFi and ONT, LongTR genotypes are highly concordant between platforms (Fig. S3).

Long TR has a consistent accuracy on both PacBio and ONT platforms that is not obtainable with SV callers like sniffles2. We have added Figure S3 which shows

cross-platform-concordance of non-TR SV calls before (41%, Fig. S3A) and after SV genotyping and sample quality filters (88%, Fig. S3B). For SVs that are spanned by TRs, raw sniffles calls have very low concordance (41%, Fig. S3C) compared to very high concordance of LongTR calls (88%, Fig. S3D). LongTR returns only one genotype per subject per TR element, thus fewer calls are made (Fig. S3D).

The challenge of calling SVs within TRs with sniffles2 in sample REACH000236 is illustrated by a representative example in Figure S3E . Long TR detects a 225 bp DEL on haplotype H1 and a 56 bp INS on H2, and the genotypes made by LongTR are identical in HiFi and ONT. From the read alignments, several deletions and insertions can be seen that vary significantly between reads. Multiple SV calls are made by sniffles2 within this region, all of which are discordant between platforms.

Studies that are done on a single HiFi platform could, in principle, call SVs within many TR elements from aligned, haplotagged reads, and such SV calls could be genetically informative in various applications of DNA fingerprinting (PMID: 3840867). In this study, however, we are more interested in accurately determining the total length of the contraction (225 bp) or expansion (56 bp) of the full repeat element than we are with defining discrete insertions and deletions within it.

Figure S3. SV and TR concordance for REACH000236 between HiFi (15x) and ONT (40x) WGS data.

(A) Non-TR SVs detected by Sniffles2 for REACH000236 before merging with other samples with 2 supporting reads show 41% concordance. (B) Non-TR SVs of REACH000236 after merging with other samples using SQ \geq 20 shows 88% concordance. (C) TR SVs detected by Sniffles2 for REACH000236 before merging with other samples with 2 supporting reads show 41% concordance. (D) TR SVs genotyped by LongTR for REACH000236 show 88% concordance. (E) An example of a TR region where alignment is noisy and introduces scattered insertions and deletions for both ONT and HiFi data. The insertions/deletions are scattered across

the TR region and SV calling needs special attention.

7. Provide individual-level statistics for the variants. What was the average number of SVs per individual? Is there a significant difference in the number of SVs per individuals between cases and controls? Is there any correlation between the number of SVs per individual and other variables such as sex and age?

While the number of SV and TR variants differs between the two platforms as was explained in response to comment 3 (Fig. S4), the number of autosomal SVs does not differ by case status (Figure S19A), sex (Figure S19B) or age (Table S20).

The following text has been added to the methods section of the manuscript:

“ We confirmed that there are no associations between the number of SV and TR variations to the case status, sex or age of the individual in the cohort (Fig. S19 , Table S20). ”

Figure S19. Number of autosomal SVs and TR-SVs stratified by case status and sex.

The number of autosomal SVs and TRs is not significantly different when stratified by case status and sex of the individuals in the cohort.

Figure S19. Number of autosomal SVs and TR-SVs stratified by case status and sex.

The number of autosomal SVs and TRs is not significantly different when stratified by case status and sex of the individuals in the cohort.

	platform	beta	SE	lower 95% CI	upper 95% CI	p-value
SVs	ONT	0.175	0.265	-0.35	0.699	0.511
SVs	HiFi	-0.051	0.586	-1.209	1.107	0.931
TR SVs	ONT	-0.202	0.368	-0.933	0.529	0.585
TR SVs	HiFi	0.073	1.124	-2.149	2.295	0.948

Table S20. Ordinary least squares model of the number of autosomal SVs and TR-SVs as a function of the age of the individuals in the cohort. We do not observe significant associations between the number of variants and age of individuals for any of ONT and HiFi platforms. The linear model is used is $variant_count \sim age$

8. Cite the previously published SV call set from short-read (SR)-WGS under the section "Comparison of SVs and TRs between LR-WGS and SR-WGS".

We have now cited "Antaki et. al. 2022" after the first sentence of the aforementioned section.

9. It would be helpful to define the criteria for "constrained genes" in the Results section. How many constrained genes were used for all downstream analysis? Provide a supplementary list.

Constrained genes are defined as genes that are loss-of-function intolerant ($pLI > 0.9$). In addition to this functional category, in response to reviewer 1 comment 18 we have also incorporated SFARI genes into our analysis, and we use the union of pLI and SFARI genes as our target gene set in Fig. 1C and 1F . This has been added to Table S2 . In total we have 4506 constrained genes with 1238 SFARI genes.

To make this clear in the manuscript we added this phrase to the results section:

“ The constrained genes are defined as the union of genes with $pLI \geq 0.9$ and SFARI genes (Table S2) ”.

10. Define "novel" dnSVs. Is it in terms of a new ASD gene, a new class of SVs, or the specific variant itself is novel?

By novel, we mean dnSVs which were detected by the LR-WGS platforms (ONT or HiFi) and not detected in our existing SR-WGS (Illumina) dataset on the same subjects.

To make the definition of novel clear, we added the following to the manuscript:

“ (detected by LR-WGS and not detected previously by SR-WGS) ”

11. The authors used a filtering threshold of at least 3 phased long reads ($AD \geq 3$) supporting the variant in the proband, but only at least 1 phased long read ($AD \geq 1$) supporting the reference allele in each parent. Why was there a difference in the read depth thresholds used?

Errors in de novo mutation calling generally consist of type 1 error in the offspring or type 2 error in the parents. Thus, confidence in a de novo SNV or SV is greatest for variants that meet stringent criteria for the quality of variant call in the offspring and relaxed criteria for the same call in both parents (PMID: 29300834). This standard for de novo mutation calling methods.

12. Clarify if the 3 dnSVs were identified in cases or controls.

The following sentence is added/modified in the manuscript to clearly mention the case status of the subjects with dnSVs:

“ The newly identified dnSVs were in two cases (REACH000426 and REACH000479) and one control (REACH000592), and all occurred within genes ”

13. Expand on the function of SH3PXD2A. What pathway is it involved in, and how does it relate to ASD pathogenesis.

Other ASD genes have similar functions in actin cytoskeleton remodeling (TRIO) and reactive oxygen signaling (UBE3A). In the paragraph describing the de novo balanced rearrangement that disrupts SH3PXD2A, we have elaborated briefly on the relevance of SH3PXD2A's molecular functions to NDDs. We do not have statistical evidence for the association of individual genes, so a lengthy discussion would not be appropriate.

“ SH3PXD2A (Tks5) is an essential gene (30) that is highly intolerant to loss-of-function variants ($pLI = 1$). SH3PXD2A is a scaffold protein involved in actin-cytoskeleton remodeling (31) and reactive oxygen species (ROS) signaling (32), pathways that have been implicated in neurodevelopmental disorders (33,34) .”

14. Axes labels for Figure 2A are missing.

Corrected

15. How many DUP-DEL-like rearrangements were identified in the entire cohort? Are the three cases presented here the only examples, or are they the most compelling examples from a larger set?

Apologies that more detail was not provided. Figure 3 reports all instances in which there is deletion nested within a duplication on the same haplotype. In addition to the 3 examples of DUP-DEL SVs shown in Figure 3, we detected a 4th DUP-DEL signature (Fig. S9) representing a duplication (chrX:155,803,260-155,987,250) and a nested deletion (chrX:155,803,824-155,983,780). However, this SV was monomorphic in our sample (allele frequency 100%). It likely does not represent a common structural polymorphism in the population, but is actually a complex DUP-DEL event that is at or close to fixation in humans. The allele represented in the grch38 reference may be a rare ancestral allele or a failure to correctly assemble the duplicated segments.

Figure S9. A monomorphic 2.7 kb insertion could be explained by a complex DUP-DEL rearrangement, and may also represent an error in the grch38 reference genome. In addition to the 3 examples of DUP-DEL SVs shown in Figure 3, we detected a 4th DUP-DEL signature representing a duplication (chrX:155,803,260-155,987,250) and a nested deletion (chrX:155,803,824-155,983,780) that reverts all but 2.7 kb of sequence of the duplication on the left, and has a neutral effect on the copy number of the gene VAMP7. However, this SV was monomorphic in our sample (allele frequency 100%). A lack of this DUP-DEL in the grch38 reference may represent a rare ancestral allele or an error in the grch38 reference in which the duplicated sequences were not correctly assembled. The same SV is represented as a 2.7 kb insertion in gnomAD v4.1 (INS_CHRX_D6524659) with frequency of 0.67. Since we have yet to find an allele that matches the reference, we suspect that this SV is not a common structural polymorphism in the population, but is actually a complex DUP-DEL event that is at or close to fixation in humans. The copy number from Illumina WGS is shown for one sample REACH000626.

16. For each of the three DUP-DEL cases described, what is the clinical status of the individual? It is a crucial piece of information to link these complex rearrangements to specific phenotypes. Robust genotype-phenotype relationships are hard to establish from a single case, but with caution, clinicians do hazard to compare diagnoses of their patients with published cases. We have clarified the clinical status of the individuals in the section below in the manuscript:

“The first example involves a large de novo rearrangement in the 8p23.1 region spanning approximately 4 Mb and flanked by two segmental duplications in an individual with ASD and aggression diagnoses. LR-WGS resolved this event as an inverted duplication (INV-DUP) of 3.8 Mb between breakpoints A and D, followed by a deletion at the junction between the two copies (Fig. 3A). The inversion signature is evident from the breakpoints observed in LR-WGS reads between the B and C junctions. A schematic illustration demonstrates how this INV-DUP-DEL rearrangement results in a coverage profile resembling a staircase (Fig. 3B). In a second case (Fig. 3C), found in a sibling control, a tandem duplication (TAN-DUP) encompasses the full-length isoforms of ZMYM2 (pLI = 0.96), GJA3 (pLI = 0.13), and GJB2 (pLI = 0), as well as the first two exons of ZMYM5 (pLI = 0). The accompanying deletion affects one of the ZMYM2 copies. The TAN-DUP-DEL architecture generates a characteristic sawtooth-like coverage pattern, illustrated in Figure 3D. The third example, found in an individual with ASD and intellectual disability, involves a TAN-DUP within the gene CDC42BPA (pLI = 1),”

Deep clinical data is not available on these subjects. Our lab has multiple projects focused on deep clinical characterization in large samples of SV carriers (e.g. PMID: 35236119, PMID: 31553903). Deep clinical characterizations are focused on a limited set of CNVs and typically involve a “gene-first” approach where the phenotyping is done in subsequent phases of the study with renewed funding. We have not yet reached that phase of our LR-WGS studies, but we hope to in the future.

17. Given the non-significant or modest p-values for many of the burden analyses, what is the statistical power of the study to detect a real effect? What sample size would be required to achieve a p < 0.05 for the burden of SVs in constrained genes, assuming the observed effect

size is real? Include a power analysis.

Indeed, while a sample of 267 long-read genomes is comparatively large by current standards, statistical power remains limited. In the current sample we are able to obtain rough estimates of the variance explained by multiple categories of genetic variation, but larger samples are needed to have good power to detect genetic associations in LR-WGS datasets.

As we expected, the sample sizes required to have good power to detect associations are larger than the current study. We performed a power analysis for SVs and TRs for specific categories of SVs and TRs, and we estimated the required number of families to detect a nominal association ($p < 0.05$, Table S15). Permutations were performed separately in PacBio and ONT dataset, which provides information on how power is affected by differences in platform and coverage. We simulated the case status of the samples within each family with weights obtained from observed effect sizes, burden values and covariate values for each functional category, while maintaining the total number of cases in the family the same. We incrementally increased the total number of samples by upsampling, and used 1000 permutations for each level of sample size. The number of samples required to achieve power=0.8 for nominal level of significance ($p < 0.05$) is provided in Table S15 .

For ONT, the number of families required to obtain statistical significance for each of the four functional categories of SV ranged from 368 to 598 families (448.5 families on average). For PacBio, the number of families ranged from 216 to 756 (531 families on average). Thus, an N of ~500 families is required to have reasonable power to detect association, and ~18% larger is required for the 4X PacBio dataset compared to the 10X ONT dataset.

SVs			ONT			PB			
variable	beta	OR	mean_power	approx_n_indiv	approx_n_fam	mean_power	approx_n_indiv	approx_n_fam	fold difference PB vs ONT
svs_fetal_brain_genes_exonic	0.34	1.405	0.982	2475	598	0.968	2134	576	0.96
svs_pil_genes_exonic	0.29	1.336	1	1513	368	0.946	794	216	0.59
svs_SFARI_high_confidence_exonic	0.83	2.293	1	1605	391	0.972	2798	756	1.93
svs_SFARI_moderate_confidence_exonic	0.38	1.462	0.932	1762	437	0.932	2129	576	1.32
Mean=					448.5			531	1.18

TRs			ONT			PB			
variable	beta	OR	mean_power	approx_n_indiv	approx_n_fam	mean_power	approx_n_indiv	approx_n_fam	fold difference PB vs ONT
trs_fetal_brain_genes_exonic	0.97	2.638	1	182	46	0.884	2013	540	11.74

Table S15. Power analysis for SVs in constrained genes, fetal brain expressed genes and SFARI genes in the high and moderate confidence categories, in exonic regions of the genome. Power analysis of TRs in fetal brain genes in the exonic regions of the genome.

We have also added the following text.

Line 394: This study provides an opportunity to quantify genetic contributions in a dataset of 267 long read genomes (117 complete trios), but statistical power is limited and larger sample sizes will be required to refine these estimates (Table S15).

Line 774: We have also performed power analyses for SV and TR burden based on permutations (Table S15). We estimate that a sample size of ~500 families are needed to have reasonable power to detect genetic associations for the functional categories of SV and TR analyzed in this study, and a sample ~18% larger is required for a dataset with 4X coverage (such as our PacBio dataset) compared to a dataset with 10X coverage (such as our ONT dataset).

18. This manuscript is a valuable contribution to the field of genomics, demonstrating the power of LR-WGS to uncover complex genetic variation. However, the overall tone feels more like a showcase of a new technology than a focused study on the genetic architecture of ASD. The narrative could be strengthened by a deeper focus on the condition. Especially because the authors have the word "autism" in the title. Some points to incorporate are listed below:

a. A more detailed clinical and demographic characterization would greatly strengthen the paper's connection to ASD. What were the precise inclusion and exclusion criteria for both the ASD cases and the unaffected controls? What are the key clinical phenotypes most frequently observed in the ASD case group?

As described in previous publications, we have added the following regarding clinical ascertainment.

“ Recruitment

Sample were collected previously as part of our project Relating Genes to Adolescent and Child mental Health (REACH) (Brandler et al. 2016) . Individuals were referred from clinical departments at Rady Children’s Hospital, including the Autism Discovery Institute, the Departments of Psychiatry, Neurology, and Speech and Occupational Therapy, and the Developmental Evaluation Clinic. Further referrals came directly through our project website. Each child included in the study received a diagnosis of ASD on the basis of an evaluation by a licensed clinician (Lord et al. 2000) . Prior to appointments, families were provided with institutional-review-board-approved consent forms and Health Insurance Portability and Accountability consent forms. DNA was obtained from 5 ml blood draws .”

The results reported here are in conjunction with the first phase of the study protocol. As mentioned above, our protocol reserves the deeper clinical characterization of subjects as part of a 2nd phase that targets specific CNVs for which large samples can be obtained.

b. The analysis of FMR1 is a technically impressive demonstration of LR-WGS capabilities, but its relevance to the primary topic of ASD is only addressed briefly at the end of the paragraph with a non-significant finding. This makes the section feel somewhat disconnected from the central narrative.

Methylation data is a key element that is provided by this technology. However, since the methylation portion of the paper was short, it did feel as if the reader was being taken on a brief detour. In response to this comment and reviewer 3 comment 9, we have expanded the methylation section of the paper to investigate imprinted genes which are impacted by LOF variants (patterns consistent with imprinting disorders). We illustrate how skewed methylation at imprinting control regions can be used to determine if a mutation impacts the expressed or silenced allele of an imprinted gene. In addition, combining our results with published data on ASD and NDD cohorts finds modest additional support implicating the imprinted gene ADNP2 in NDDs but not in SPARK. Further studies of ADNP2 in clinical cohorts are needed to determine clinical phenotypes associated with ADNP2, but this example does highlight how one can combine the phasing of SVs and methylation to find signatures consistent with an imprinting disorder. To highlight more clearly methylation of imprinting control region of ADNP2 in the paper, we have sequenced this trio to 30X using the new PacBio Revio instrument at UCSD. Limited evidence for imprinting disorders

Phased methylation data from LR-WGS can also be used to determine the imprinting status of an allele which can facilitate the diagnosis of an imprinting disorder such as Prader–Willi syndrome (PWS) or Angelman syndrome (AS) ^{47,48}. As expected, several known imprinting control regions (ICRs) showed strongly skewed methylation in our dataset, for instance the PWS/AS CNV region (Fig. S10A-B), as well as GRB10 (Fig. S10C) and GNAS (Fig. S10D). We examined the SVs and SNVs detected in our cohort for a pattern consistent with an imprinting disorder ⁴⁹. Exonic deletions or protein-truncating SNVs were intersected with a database of imprinted genes ⁵⁰. LoF variants in 4 putatively imprinted genes were identified (ANO1, ERAP2, ZNF396, ADNP2), of which one (ADNP2) had an ICR that could be confirmed to have skewed methylation in the phased methylation data from this study. In one family, we found a maternally-inherited deletion of a maternally-expressed gene ADNP2 (Fig. 4A). Skewed

methylation of the ICR (chr18:80,159,520-80,160,720) in the parents confirmed that the deletion (chr18:80,045,344-80,157,432) was present on the mother's inactive (paternal) allele (Fig. 4B-C). This proband was also determined to have XYY syndrome, a known contributor to

autism, so at most this variant could be a potential genetic modifier.

ADNP2 (Activity-Dependent Neuroprotective Protein 2) encodes a homeobox-containing protein expressed in the brain and predicted to act in transcriptional regulation and neuronal function ⁵¹. It is a paralog of ADNP, a gene that is associated in ASD and neurodevelopmental disorders ⁵². A recent genetic study found weak evidence for associations of ADNP2 with developmental delay ⁴ but did not detect association with ASD. Further investigation of loss of function variants in the SPARK dataset did not find evidence implicating ADNP2 or the adjacent maternally-expressed gene PARD6G (Table S9). This example highlights how one can combine the phasing of SVs and methylation to find signatures consistent with an imprinting disorder; however there remains weak statistical association of ADNP2 with neurodevelopmental disorders.

Figure 4. Deletion of the imprinted gene ADNP2. (A) A maternally-inherited deletion of the maternally-expressed gene ADNP2 was detected in proband REACH000293 (chr18:80,045,344-80,157,432). The deletion hat spans the full ADNP2 gene and non-coding segments of the adjacent genes RBFA (3'UTR), PARD6G (3'UTR), and RBFADN (lncRNA). Linked reads highlight the breakpoints of the deletion and color represents the haplotype (Red: maternal, blue: paternal, gray unphased). (B) Methylation of CpG sites (red=methylated, blue=unmethylated) is shown for the imprinting control region. In this case, the methylated (red) haplotype is the maternal (expressed) allele. (C) Based on phased methylation data, the deletion is on the maternal (expressed) allele in the proband and is present on the paternal (inactive) allele in the mother. The bar heights and error bars correspond to median and median absolute deviation (MAD) of the data.

Figure S10. LR-WGS reveals methylation bias in maternal and paternal haplotypes in Imprinted SFARI genes.

(A-D) LR-WGS methylation signature in four imprinted SFARI genes: MAGEL2, SNRPN, GRB10 and GNAS (Jima et al. 2022; Akbari et al. 2024) for REACH000236 with ONT data.

c. The author uses a constrained gene list for downstream analysis, but does not explicitly incorporate the widely-used SFARI gene list. A burden test on SFARI genes would provide a more targeted, disease-specific analysis. What % of the constrained genes are ASD genes? Our manuscript is intended to illustrate what can be assessed with LR-WGS and to highlight analyses that are simply not done in traditional SV, WES or GWAS workflows. In that sense, it is indeed a showcase of a different analytic framework, one that interrogates a broader range of structural and functional genomic features. At the same time, we agree with Reviewer 1 that the relevance of the structural variants (SVs) and genes to ASD should be more explicitly emphasized throughout the manuscript.

As suggested, we have now incorporated the known autism genes (SFARI genes) into our downstream analyses. Specifically, we used the combined set of SFARI genes (1,238 genes total: 428 syndromic or category 1 [high confidence] genes and 810 category 2 or 3 [strong candidate or suggestive evidence] genes) to test the association of exonic SVs and tandem repeats (TRs). We then added SFARI gene-specific burden estimates and r^2 values to the association results in Figure 6.

We have also clarified and expanded our definition of constrained genes. The constrained gene set, now provided in Table S2, is defined as the union of genes with $pLI \geq 0.9$ and SFARI genes. In total, this set comprises 4,506 constrained genes, of which 1,238 (27%) are ASD-associated SFARI genes. This makes the ASD relevance of the constrained gene set more explicit. In the course of revising these analyses, we found that de novo missense variants were inadvertently omitted from the R^2 estimates of all rare variants combined in Figure 6. We have corrected this, recalculated the SNV burden to include de novo missense variants, and updated the r^2 and h^2 estimates accordingly. In addition, we have added burden analysis of the SFARI gene-set for both SVs and SNVs to Figure 6. Including these additional terms in the regression models results in very minor changes to the r^2 and h^2 estimates. The table below compares the new results with the old.

Category	R2 (Original)	R2 (add dnMis)	R2 (add dnMis & SFARI genes)	h2 (Original)	h2 (add dnMis)	h2 (add dnMis & SFARI genes)
SVs (pLI\geq0.9)	0.0558	0.05439	0.05709	0.03414	0.03325	0.03495
TRs (fetal brain)	0.0261	0.03043	0.03159	0.01573	0.01838	0.01909
SVs + TRs	0.06627	0.06744	0.07291	0.04076	0.0415	0.045
SNVs	0.02974	0.0476	0.04619	0.01796	0.029	0.02812
Rare variants combined	0.09884	0.11542	0.11651	0.06182	0.07281	0.07354
PGS (ASD,SCZ,EDU)	0.02624	0.02832	0.02619	0.01582	0.01709	0.01579
Rare variants + PGS	0.12034	0.13653	0.13759	0.07611	0.08708	0.08781

Reviewer #2 : The manuscript by Milad Mortazavi and colleagues describes the use of long-read sequencing to characterize genetic variation in autism spectrum disorder (ASD). The genomes of 267 individuals from 63 families have been sequenced using either PacBio or Oxford Nanopore sequencing. Variant call sets are compared to previously generated short-read results. Specific complex SVs are described in greater detail. Methylation analysis is performed focusing on FMR1, showing that gray-zone alleles exhibit increased methylation. Overall, the conclusion of the paper is that rare variants, and especially SVs, explain 6.2% of ASD heritability in this cohort.

This is a well defined cohort and long-read sequencing of such cohorts are of general interest, making the general theme of the paper interesting and timely. However, there are also a number of issues with this study that will need to be addressed in order to publish the manuscript in a high impact journal.

1) The introduction does not really bring up any of the studies that have been done previously to show the benefits of long-read sequencing in comparison to short-read sequencing, with regard the improved SV calling. Since it is highly relevant to the paper, it should be added, with proper references to the most prominent studies.

We Thank the reviewer for this suggestion. We have added a more detailed summary of long read sequencing capabilities and advantages to the introduction:

“ A portion of the genetic contribution to ASD could lie within genomic variation that is poorly captured by current short-read whole genome sequencing (SR-WGS), particularly rare SVs greater than 50 base pairs in length (11) and variable number tandem repeats (VNTRs) that cannot be fully resolved with short reads (12). LR-WGS has demonstrated substantial advantages over SR-WGS in the detection of SVs and TRs. The longer read lengths, ranging in size from 5 kb to 500 kb (13), enable direct resolution of repetitive and structurally complex regions that are often collapsed or misassembled in SR-WGS (14,15), and allows the assembly of phased SV haplotypes (16). LR-WGS technologies were instrumental in generating telomere-to-telomere (T2T) genome assemblies that have closed long-standing gaps in the human reference genome GRCh38 (17,18). These advances refine our understanding of

genome architecture and also enhance the interpretation of pathogenic variants in previously uncharacterized loci.

In addition to improved variant detection and assembly, LR-WGS can accurately call base-modifications, particularly DNA methylation, without requiring separate bisulfite or enzymatic conversion assays. This enables the simultaneous profiling of genetic and epigenetic variation from the same genome sequence, facilitating integrative analyses that link rare variants to methylation epigenatures and disease mechanisms (19–21). Collectively, these features enable more comprehensive genome analysis and variant interpretation. ”

2) The sequence coverage is very low for most of these individuals, and the sequence data is generated by "older" generations of PacBio (Sequel IIe) and Oxford Nanopore (R9) flow cells. Especially the low coverage is really the main issue with this study. While the ability to detect SVs is still decent with limited coverage of long reads, it is too low here for many individuals to really call variants reliably i large parts of the genome. Low coverage will also reflect poorly in large CNV regions where breakpoints are in long repeats, i.e. where detection is based on fluctuations in read counts/coverage. This also explains why so many short-read only SVs were detected. Compared to other long-read benchmark studies the number of SVs identified is very low, as a result of low sequence coverage and possibly in combinations with calling parameters. Mean coverage of ONT was 10X and mean coverage of HiFi as 4X, and we provide a detailed evaluation of SV calling accuracy at varying levels of coverage. This study reports diverse forms of structural and functional variation that can be captured in hundreds of genomes. Genetic analysis is well controlled for effects of platform and coverage, and the claims being made are well supported. We further address this point above and in response to reviewer 3 comment 1B. We agree with the reviewer that deeper sequencing on a larger scale is needed to move beyond the results that we are reporting here.

The standard that reviewer 2 is referring to in terms of sequencing coverage and cost is possible with new instruments that have been widely available for less than 2 years. The PacBio Revio instrument can yield >30X coverage of a sample from a single flow cell at a cost ~\$2,000 per sample. The Revio was first announced in 2022, but as the reviewers may recall, shipments of the actual instruments did not come until later. UCSD received its Revio instrument in 2024. Publications using the Revio instrument, such as our recent AJP paper (Mortazavi et al. 2025), are being done on a small scale. In revising this manuscript we also sequenced select samples to 30X with the revio (Fig. 4). Studies that are being published today that consist of hundreds of genomes will be datasets that were collected by early adopters over the last several years using slightly older technologies and standards for coverage .

Overall, it is questionable whether this level of coverage will give very useful new insights beyond the off finding of some specific individuals with interesting rearrangements or methylation profiles.

The single biggest technical factor that determines what kind of analysis is possible in a dataset is the sample size . If we had taken this reviewer comment to its logical conclusion, and we had insisted on sequencing all samples to 30X (resulting in a total N of 50 instead of 243), then this paper would definitely have been series of anecdotes. Fig 2 would likely be reporting one novel de novo coding variant, Fig 3 would be describing a single anecdotal example of a DUP-DEL event. Fig. 5 (effects of CGG repeat expansions on methylation) would have detected only 1 or 2 gray zone expansions and would simply not have been possible. The analysis of imprinted genes would have produced only the descriptive results in Fig. S10 (there likely would be no Fig. 4). Lastly, sample size is critical for genetic association. Even the current sample is underpowered, but we are able to make rough quantifications of genetic contributions when multiple categories of rare variants are combined. I think it is plain to see that there was value in

increasing the sample size. While lower coverage is a technical issue that must be addressed throughout the analysis, to say that this paper is a series of “off findings” is a mischaracterization. This paper consists of a series of analysis that would not have been possible in a sample size <100.

3) *The authors discuss long-read sequencing as if it is one method, while they have in fact used two different methods for long-read sequencing. These methods will find different sets of SVs and SNPs, have slightly different strengths and weaknesses in estimating VNTR length, different algorithms for 5mC calling etc.*

We have added detailed analyses of strengths and weaknesses of each platform in response to reviewer 1 comment 1 and reviewer 3 comment 1C.

4) *Since read-length may vary greatly for long-read methods depending on how the DNA was treated and how libraries were constructed, it should be stated in the text what the average read-length and distribution are for each of the long-read platforms. Now these quite informative numbers are only found in Supplementary Table 1. The mean read length for some of the nanopore is actually quite low and will affect the ability to call variants.*

As is requested by the reviewer, we have added Figure S1 illustrating the read length metrics and sequencing coverage distribution to the manuscript. The average read length in the ONT data (~5,600 bp) is lower than that of the HiFi data (~11,300 bp). However, good accuracy is obtained for the ONT data. This is evident from the ROC curves showing the sensitivity and FDR of SV calling pipeline for both platforms as a function of sequencing coverage (Figure S2) for REACH000236, with average read length of 6,400 bp and 10,800 bp for ONT and HiFi data respectively.

To clarify the read length statistics of each platform, we have added a sentence indicating the mean read lengths in the first paragraph of the section titled: “ Calling SVs and TRs from LR-WGS ”:

“, with mean read length of ~5600 bp for ONT and 11,300 bp for HiFi sequencing (Fig. S1). ”

Figure S1. Read length and coverage distribution of subjects in the cohort.

(A) Read length distribution stratified by platform. (B) Coverage distribution of subjects in the cohort stratified by platform.

5) *The first paragraph in the results section is slightly confusing in terms of what individuals/families were sequenced. It seems to be a mix of incomplete families, trios and quads (and possibly larger families?), but this is impossible to decipher from the way it is currently written. It is clear from Table S1, but should be written more clearly also in the text.*

Like other family cohorts such as SPARK, the REACH study is a mixture of family structures

(trios, quads, quintets and a few duos) and the vast majority of families have at least one case trio (case with mother and father). Apologies for any confusion. We have modified the first sentence to clarify the number of individuals and their subcategories as:

“ We sequenced 267 individuals (Table S1); 243 of whom are within complete trios from 63 families (some families have more than one offspring). This includes 117 offspring (76 cases, 41 unaffected controls, 74 males, 43 females), and 126 parents. Of the total, 158 individuals were sequenced using the PacBio HiFi platform (Sequel IIe) and 109 using Oxford Nanopore Technologies (GridION), with mean read length of ~5600 bp for ONT and 11,300 bp for HiFi sequencing (Fig. S1). ”

6) *The end of the first paragraph and start of second paragraph in the section about de novo SVs are unclear in terms of number of de novo events identified and confirmed. 65 candidate dnSVs were identified and SR-WGS was used for validation and genotyping. But in the next sentence it says 6 were unique to SR-WGS - clearly not referring back to the previous sentence. Some clarification is required for the reader to actually follow the reasoning here.*

By 65 candidate dnSVs we mean those detected with the stringent filtering from both LR-WGS and by SR-WGS previously, Some of which overlap between the two platforms. After cross-validating dnSVs across LR-WGS and SR-WGS platforms we obtained 15 dnSVs from both platforms in our cohort in total (50 putative de novos were invalidated). It is customary for de novo SV calls to have a validation rate of <25% (Brandler et al. 2016) . To make this procedure clear we have changed the text as follows:

“ This stringent filtering of long read SV calls in addition to the SVs which were previously detected from SR-WGS (Brandler et al. 2018) yielded in total 65 candidate dnSVs. To validate the putative dnSVs detected from LR-WGS, we performed orthogonal genotyping of putative LR-WGS dnSVs using SR-WGS data (Methods). In total, we confirmed 15 dnSVs, 3 unique to LR-WGS, 6 unique to SR-WGS, and 6 detected by both, in 11 ASD cases and 3 unaffected controls (one case had two dnSVs) (Table S5). ”

7) *The calling and validation of dnSVs is also a bit unclear in the methods section. It is stated that high confidence de novo SVs requires allele depth >3 in the offspring, and zero AD in both parents, each of whom must have 1 or more phased long read supporting the reference allele in both haplotypes. With this definition it is unclear how a de novo duplication can be called? Clearly a duplication allele can not have 0 read depth in both parents as one of the alleles must be the one that is duplicated?*

In rare variant detection nomenclature, allele depth refers to the number of reads supporting the ALT allele (not both). To make this more clear, we added “alternative” before “allele depth” in the methods section. A sample can have zero alternative allele depth (AD) for a variant, where all reads support the reference (therefore a non-zero coverage depth). In other words, the parents can both have AD=0 (no reads supporting the duplication), while having multiple reads supporting the reference in that region (therefore having a high coverage in that region). We require at least 3 reads supporting the putative dnSV in an offspring and an absence of reads supporting the allele in both parents. We also require there be at least one phased read in each haplotype of both parents to make sure we do not fail to observe a haplotype in the parents.

In order to make this more clear, we changed the sentence in the methods section as:

“ High-confidence de novo SVs were defined as those with alternative (ALT) allele depth (AD) ≥ 3 in the offspring, and zero ALT AD in both parents, each of whom must have ≥ 1 phased long read supporting the reference (REF) allele on both haplotypes. ”

8) *It also seems strict to require phasing in the long-read data since phasing often falls apart in rearrangement regions, especially if surrounded by long repeats, which is often the case for SV*

regions. The definition that both parents must have one or more phased long read supporting the reference allele on both haplotypes also seems like it would exclude certain events, e.g. where a parent carrying an inversion haplotype has an off-spring with de novo del or dup in that region (a mechanism well established in e.g. the Williams region).

The reviewers point is well taken that, if there is good coverage but no phasing, then we would be excluding a de novo SV unnecessarily. In response to this comment (and a similar comment 3 from reviewer #3), we have fully relaxed these conditions for all 211 SVs in constrained genes and SFARI autism genes (Fig. 1C), 47% of which did not meet our strict criteria originally. We reviewed these variants manually in both LR-WGS and SR-WGS in every trio to determine if the variant is inherited or de novo. This analysis did not identify any new de novo variants, but it certainly could have. We have added this to the paper.

One additional point: in most instances, a lack of phasing simply reflects a lack of informative heterozygous SNPs. Scenarios like the one described by the reviewer (a de novo deletion in an offspring that occurs on an inverted haplotype in mom) are often phased correctly. Phasing of reads by WhatsHap does not require that all SNPs be contiguously aligned along the full length of the read. If an inversion on haplotype H2 splits a read into two alignments. The two halves will generally still match H2 and will be correctly haplotagged.

9) Were there no LR-WGS dnSVs called in regions where SR-WGS could not be used for validation due to issues with mapping? What happened to such candidate dnSVs in this study?

None of the 65 candidate dnSVs were disregarded due to insufficient mappability in SR-WGS. All of the 65 candidate dnSVs had >15X SR-WGS coverage in all members of the 60-65 trios.

10) It is unclear for the rearrangements described in Figure 3 what evidence there is for the rearrangement architecture. Are there breakpoint spanning reads that confirm the order and orientation of sequences? As an example (but valid for all complex SVs in figure 3) in Figure 3b, is there evidence that the deletion within the tandem dup is upstream of the complete copy?

We originally felt that providing further detail in Figure 3 on the signatures that are evident in the raw reads (along with gene tracks and diagrams of the DUP and DEL events) would make the figure dense and complex. We have now added Figure S7, which provides read-level evidence of complex rearrangement using junction-spanning reads and their alignments to the reference genome. This should give the necessary details on the distinct signatures of deletions and duplications (Figure S7 B-C) as well as inversions (Figure S7 A) in complex SVs that are seen from long reads and their alignments to the reference genome. We reference Figure S7 here and in subsequent questions on this topic.

When there are reads that span both the deletion and at least one of the duplication boundaries (Figure S9) it is possible to determine which (right or left) copy contains the deletion. In another recent publication of ours (Mortazavi et al. 2025), this is how we determined the functional consequence of a complex SV (RFX3 haploinsufficiency). If there are no such reads, then we cannot unequivocally determine which copy has the deletion. Fig. 3A and Fig. S7A describe a large INV-DUP-DEL at 8p23 in which the deletion spans the duplication junction between breakpoint B in one copy and breakpoint C in the other copy. In this case, there is no read that spans both DUP and DEL junctions, so we are not able to determine which copy contains the B breakpoint and which contains the C breakpoint. By default we place in the B breakpoint in the left copy. We have now added this information to the figure legend.

Figures 3C and 3E are also examples of TAN-DUP-DEL where we do not have a read that anchors the deletion to the left or the right. The long-read support for tandem duplications and deletions are depicted in Figures S7 B-C. Since we do not know which copy contains the deletion, we arbitrarily place the deletion on the left copy (and this is indicated in the figure legend). For a common TAN-DUP-DEL complex SV described in Figure S9, there are reads

that span the deletion and left boundary of the duplication, thus we can infer that the deletion is on the left copy.

Figure S7. Split long reads passing deletion/duplication junctions support presence of complex SV breakpoints.

(A) Corresponding to the complex SV in Figure 3A. Read 1 and 2 passing the inversion junction generate split alignments between B-C in the reference genome. (B) Corresponding to the complex SV in Figure 3C. Reads spanning the deletion and duplication junctions generate split alignments between B-C and A-D in the reference genome. (C) Corresponds to the complex SV in Figure 3E. Reads spanning the deletion and duplication junctions generate split alignments between B-C and A-D in the reference genome.

10B. Are there breakpoint reads that can be used to exclude that one copy is inverted?

Yes a tandem duplication has a distinct pattern of split read alignments that is distinguishable from an inverted duplication. The inversion can be inferred from junction sequence of the B-to-C breakpoint. The long-reads supporting the inversion between the B-C junctions are shown in Figure S7A. It is a fortunate coincidence that the DEL spans both copies and has completely deleted the cluster of segmental duplications at the duplication junction. If the DEL had been nested within a single copy of the duplication, it would have been a challenge to correctly assemble the highly-repetitive duplication junction and to infer the orientation of the two copies in Figure 3A-B.

10C. Is there any evidence that this is a TAN-DUP-DEL rather than two smaller duplication events (i.e. instead of initial tandem dup followed by del, there would be two smaller duplications events (and no deletion))?

Yes. If these were two smaller duplications, there ought to be evidence of 2 separate duplication junctions and there would be no deletion junction. Instead, as shown in Figure S7, the duplication-junction spanning reads connect the A-D junctions. This is evidence of having a large duplication with a smaller deletion inside one of the copies instead of having two separate duplications without a deletion in between. Only the short read (coverage) data is misleading. The long read data is not.

10D. Are there details around the breakpoints that may help elucidate the mechanism (e.g. inserted bases, microhomology etc)?

We have investigated breakpoints of deletions, duplications and inversions in Figure 3 to determine the mechanism for creation of the complex SVs. The sequence details of the breakpoints and their alignment to the reference are summarized in the supplementary material. The following brief description can be found in the main text.

“Analysis of the breakpoint sequences of the variants in Figure 3 find evidence for a mixture of mutational mechanisms including non-allelic homologous recombination (NAHR), microhomology mediated end joining (MMEJ) and non-homologous end joining (NHEJ) (Supplemental Material). In these examples, the duplication and the nested deletion of the same DUP-DEL variant often appear to occur by different repair mechanisms.”

In total we found signatures of non-allelic homologous recombination (NAHR, Fig. 3A duplication breakpoint), microhomology-mediated end joining (MMEJ, Fig. 3C deletion and 3E duplication), and non-homologous end joining (NHEJ) with short templated insertion (Fig. 3C duplication).

11) For the FMR1 analysis, it is unclear how many females that actually pass the requirement of having at least 3 phased reads per haplotype, and how many end up in each of the three groups (gray-zone, short, intermediate). It would be helpful if the numbers are stated in the text.

22 females had the necessary coverage of both haplotypes (N=5 gray-zone, N=6 short allele and N=11 intermediate). The text is modified as follows:

“ We extended this analysis to all female subjects with at least three phased reads per haplotype and categorized them into three groups based on CGG repeat length: (1) those with at least one gray-zone allele (≥ 35 repeats, N = 5), (2) those with at least one short allele (≤ 25 repeats, N = 6), and (3) those with both alleles in the intermediate range (26–34 repeats, N = 11). ”

Reviewer #3 : Mortazavi et al. present a study in which they sequenced 243 individuals from 67 families using long-read sequencing technologies. The authors perform multiple analyses,

including calling structural variants (SVs) and tandem repeats (TRs), comparing long-read derived SVs with previously generated short-read SV calls, detecting potentially relevant tandem variants, identifying SV signatures, assessing methylation near FMR1, and evaluating the burden of variants in autism spectrum disorder (ASD). The manuscript leverages a well-established cohort and applies advanced sequencing technologies; however, there are several important limitations and methodological concerns that warrant discussion.

Strengths

This study benefits from a uniquely valuable cohort: families with ASD that have been extensively characterized over many years in the Sebat Lab. Applying long-read sequencing to such a resourceful dataset represents a meaningful step toward discovery.

The integration of diverse analyses spanning SV calling, TR detection, methylation analysis, and burden testing provides a broad perspective on the potential contributions of structural variants and tandem repeat variants to ASD.

The paper addresses a timely and compelling topic.

Major Concerns

1. Sequencing Depth and Platform Heterogeneity

Question 1 is in multiple parts, which we will address one at a time. We gave strong consideration to both coverage and platform heterogeneity. However, these issues were not described in depth in the original submission. We provide a more detailed explanation of how these technical confounders were addressed. The following downstream steps are required to get accurate results from multiple sequencing platforms with varying levels of coverage.

1. Optimize downstream QC to produce call sets with comparable quality despite differences in the total of variation that may be captured on a given platform.

2. Control for the effects of coverage in statistical models

3. Stratification by platform and combining platforms by meta-analysis

1A. I would expect at least 30x coverage for PacBio to be a standard coverage with high coverage being substantially higher (e.g., 60x).

Data collection for this project concluded in 2023. During this project period the reviewer may recall, projects by large consortia that sequenced genomes to 30X coverage on the Sequel II platform had sample sizes of ~30 (PMID 33632895). The 30X “standard” that reviewer 3 referring to was not possible to do at scale with R01 level funding, thus our NIH grant was funded to sequence hundreds of genomes at 4X-10X. There is a clear rationale for our design choice. As was originally proposed in our NIH grant, this study managed a tradeoff between sequencing depth and sample size in order to achieve sample size >200, and we provide extensive evaluation of the performance of our methods at low coverage. That this choice was justified should be clear from the results. Many of the analyses performed here would not be possible in a sample of 30 genomes.

1B. The most significant limitation is the shallow sequencing depth. Reported average coverage (Table S1) is only 3.8x for PacBio and 10.7x for ONT. Such low coverage substantially reduces variant detection sensitivity and raises concerns about both false negatives and false positives. Figures S1 and S2 further highlight the suboptimal performance associated with these specific depth levels. While the authors attempt to adjust for coverage by including depth as a covariate, such modeling cannot fully substitute for adequate raw coverage.

With new technologies, it is generally the case that sequencing at high coverage in large samples is cost prohibitive. When applying a new technology to disease cohorts, it is customary therefore to seek a balance between coverage and sample size and to optimize workflows accordingly. As my lab has demonstrated extensively in the last 20 years, pathogenic protein-coding variation contributing to psychiatric disorders can be detected with low resolution

platforms such as microarrays (Shanta) and likewise WGS can perform well at low coverage <https://jmg.bmj.com/content/55/11/735.abstract> . Power in this study would NOT be increased if we cut the sample sizes in half and doubled the coverage. But the reviewer is correct that this tradeoff is something that must be managed carefully in designing a study.

In response to reviewer 1, comment 17, we calculated the number of families required to obtain significant associations for SV burden in various genomic categories. The power analysis results showed that given the current level of coverage in the two platforms, the PacBio cohort needed 18% more families, on average, compared to the ONT cohort, despite significantly lower coverage in the PacBio samples. It's evident that increasing the number of samples two fold has a more significant effect on the power than doubling the coverage.

As described in our methods in the original submission, we carried out early pilot studies to establish the coverage required for each sequencing platform to achieve reasonable accuracy (AUC ~ 0.8).

1. We created a truth set of SV calls that have minimal dependence on sequencing depth and quality filtering parameters

Using higher coverage data HiFi and ONT platforms, we defined the truth set as the intersecting calls between the 2 platforms. This gave us SV calls that were representative of those detectable by long reads, and were not heavily biased toward specific loci that are well represented in low coverage data.

2. We then evaluated the sensitivity and false discovery rate (FDR) of each platform as a function of sample quality filter and sequencing coverage. With our $SQ \geq 20$ filter, sensitivity and FDR of our variant calling pipeline was ~71% and 11% respectively for HiFi 5X and ~78% and 15% respectively for ONT 10X. Thus, these represent the minimal levels of coverage required to have accuracy comparable to what is routinely seen in large scale studies of rare CNVs. While reviewer 3 would prefer to see the kind of coverage that's now possible at \$2000/sample on the Revio platform, the choices that we made in this project were well justified and were adequate for the goals of this study. Increasing the PacBio and ONT coverage does improve accuracy from AUC ~ 0.80-0.81 at the current level to AUC ~ 0.86-0.88 if we doubled the PacBio and ONT coverage to 10X and 20X respectively. However this, in my view, does not justify cutting our sample size in half.

Figure S2. Benchmarking snoopSV (an in-house genotyping method) to detect allele depth (AD) and assign quality metrics to SVs.

(A) Sensitivity vs. FDR for HiFi data. (B) Sensitivity vs. FDR for ONT data. (C) True Positive Rate (TPR) versus False Positive Rate (FPR) for HiFi data. (D) True Positive Rate (TPR) versus False Positive Rate (FPR) for ONT data. Area Under the Curve (AUC) for each curve is annotated in the figures. TPR: True Positive Rate. FPR: False Positive Rate. FDR: False Discovery Rate.

A subsequent comment #3 from Reviewer 3 specifically asks about sensitivity for detecting de novo SVs when stringency of filters is further increased, which we address later in response to that comment.

1C. Compounding this issue, different samples were sequenced on two distinct platforms with very different error profiles and read characteristics. Mixing PacBio and ONT data introduces additional variability, making it difficult to distinguish biological signal from technical noise. For example, a variant absent from a PacBio sample may be missed due to very low coverage, while ONT-specific artifacts could lead to spurious calls. Are the covariate models robust enough to adjust for both within-platform and cross-platform biases?

See also response to Reviewer 1 comment 2. Platform heterogeneity is a well-known technical confounder that we are intimately familiar with and have specifically addressed in this study and others. These are NOT insurmountable confounders when one applies standard statistical

genetic approaches that are used in GWAS and other studies that combine multiple datasets from different platforms. First, SV burden within functional categories is analyzed WITHIN platform. Coverage is a covariate in the model. Platform is not. Second, results from the 2 platforms are then combined by meta-analysis. As we have demonstrated previously, this approach is effective for eliminating spurious associations due to platform heterogeneity (Shanta et al).

Statistical models : coverage has a significant influence on the burden of variants detected in a sample, thus it is essential to control for coverage in statistical burden tests. Our statistical models test for SV burden within specific functional categories and account for the effects of coverage and the genome-wide burden of SVs detected.

The full model tests SV burden within category and controls for genome-wide coverage, and genome-wide SV burden. The model also includes other genetic predictor categories (e.g. de novo SNV burden, polygenic risk) as well as other confounders (sex and ancestry principal components).

Phenotype ~ category_burden + sex + coverage + count_dnlof_inhlof_dnmis + PRS_ASD_Z + genome_wide_burden + PC1 + PC2 + PC3 + PC4 + PC5 + PC6 + PC7 + PC8 + PC9 + PC10 + strata(fid)

The null model includes everything but category SV burden:

Phenotype ~ sex + coverage + count_dnlof_inhlof_dnmis + PRS_ASD_Z + genome_wide_burden + PC1 + PC2 + PC3 + PC4 + PC5 + PC6 + PC7 + PC8 + PC9 + PC10 + strata(fid)

Using ANOVA to find the significance of the burden variable (p-value):

```
ano <- anova(null_model, full_model, test='LRT')
```

Stratification by platform : Statistical tests are performed separately within each platform. See response to reviewer 1 comment 2.

Stratification of associations by size, genes and autism genes . Last but not least, spurious results that are driven by low coverage should broadly affect many types of SVs. We would expect that spurious results would similarly affect large and small SVs, and genic and intergenic SVs, and associations would not be concentrated in autism genes. Prompted by reviewer 1 comment 18, we have stratified our associations by functional consequence (intergenic, intronic, exonic), functional constraint (pLI) and autism genes (SFARI genes) (Fig. 6). Prompted by this comment by reviewer 3, we have stratified our association data by functional consequence (intergenic, intronic, exonic) and size (Fig. S14). We observe no significant associations for intronic and intergenic SVs across all size ranges, and the only association signal that is evident consists of coding SVs (Fig. S14) and again association signal is concentrated within coding regions of constrained genes and autism genes. (Fig. 6).

Figure S14. SV burden association with ASD case status stratified by SV size and functional consequence.

No association is observed in the intronic and intergenic categories, and the only association observed is in the large and exonic SVs.

1D. the study does not include a direct within-sample comparison of PacBio versus ONT even for the one sample described as "high coverage" (40x ONT, 15x PacBio), which still falls short of standard high coverage for SV analysis.

Why was a direct PacBio-ONT comparison on the "high-coverage" sample not included? How might platform-specific error modes (e.g., ONT homopolymer-associated errors) bias detection of certain SV classes?

We have now added a within-sample comparison of SVs and TRs (>50 bp expansion/contraction) between HiFi and ONT platforms for the high-coverage sample (REACH000236) as Figure S3. Panel A shows 41% concordance between the platforms when we detect SVs using Sniffles2 and 2 supporting reads. However, when we go through the pipeline and merge SV calls across the cohort, genotype them and select variants with $SQ \geq 20$, we observe 88% concordance between the two platforms for REACH000236 (Fig. S3B). Similarly, TR-SVs detected by Sniffles2 and 2 supporting reads for the same sample show 41% concordance (Fig. S3C), while when we use LongTR to detect high quality SVs in TR regions we obtain 88% concordance (Fig. S3D). This shows that our SV detection pipeline (both in TR and non-TR regions) are not significantly affected by utilizing two LR-WGS platforms. Despite well-characterized differences in small-scale variations such as SNVs and indels between the ONT and HiFi platforms, large-scale SV detection is not significantly impacted by the biases in platform-specific error modes.

We have added this to the manuscript:

"The concordance of the HiFi and ONT platforms for the SV and TR call sets of REACH000236

is evaluated in Figure S3 . The concordance for both SV and TR call sets was 88% using our SV and TR detection pipelines. ”

2. Choice of Variant Calling Tools

The study appears not to have used the most widely adopted long-read SV/CNV callers. For PacBio HiFi data, callers such as pbsv and hifcnv are standard. Other popular tools such as cuteSV and SVIM were also not considered in the analyses. Why were PacBio-specific tools (e.g., pbsv, hifcnv) not used? Was this a deliberate methodological choice? Were cuteSV or SVIM considered? Were existing benchmarking studies (e.g., GIAB truth sets) used to guide tool selection? Could omission of these callers disproportionately affect detection of certain SV classes (e.g., tandem repeats, insertions, large deletions)?

Tools for SV detection from long-read data have evolved rapidly in recent years. However, maintaining version control and consistency in raw data processing pipelines is essential when conducting large-scale genome sequencing projects that span several years. Most of the methods mentioned were evaluated during the course of this project. The SV callset analyzed in this study was finalized prior to the initial release of hifcnv in 2023.

During the early years of the project (2017–2021), we periodically benchmarked newly published tools against our established pipeline. We performed head-to-head comparisons of Sniffles with pbsv, cuteSV, and SVIM at multiple time points. Although pbsv performs well on PacBio data, it is heavily tuned to the specific error profiles of that platform and not optimal for

multi-platform datasets, which were part of our broader project. *cuteSV* and *SVIM* also demonstrated comparable sensitivity and precision for certain variant classes, but none offered features essential for our large-scale multi-sample analysis.

A major advantage of *Sniffles2* was its introduction of the *.snf* file format, which stores detailed breakpoint and alignment information for each sample. This file format enables joint genotyping across multiple genomes, producing a unified SV genotype matrix similar in concept to GATK's gVCF workflow for small variants. At the time of analysis, this capability was unique to *Sniffles2* and not available in *pbsv*, *cuteSV*, or *SVIM*.

Finally, we deliberately avoided an ensemble approach (merging multiple SV callsets) because integrating calls from different algorithms often introduces additional complexity and error. For large CNVs detected in short-read data, an ensemble approach can be effective since breakpoints are approximate. However, for smaller (50–100 bp) insertions and deletions detected at single-base resolution, “consensus” merging of overlapping calls can paradoxically reduce accuracy relative to the best-performing individual method. For this reason, we opted to use a single, consistent long-read caller—*Sniffles2*—across all samples.

The choice of TR genotyper: *LongTR* was based on its ability to genotype a general list of pre-defined TR regions genome-wide (STRs and VNTRs) scalably and efficiently for both HiFi and ONT data. These capabilities enabled us to unbiasedly genotype the two platforms across our cohort. The concordance between TR genotypes of a high-coverage individual is discussed in the previous comment. Other available TR genotyping tools exist such as *TRGT* and *Straglr*. However, *TRGT*'s parameters are tuned for highly accurate HiFi data and struggles with high error rate ONT data. *Straglr* does not require a predefined TR list and can detect *de novo* TR variations, however, it is computationally very expensive for hundreds of thousands of genome-wide TR loci and large cohorts. *LongTR* was the unbiased and most efficient tool to genotype both HiFi and ONT data for genome-wide TR regions in our cohort.

3. Validation of De Novo Variants

The criteria for defining de novo variants appear underdeveloped. With PacBio coverage averaging only 3.8x, a threshold of allele depth >3 risks missing genuine de novo events. Moreover, it is not clear what orthogonal validation strategies were utilized (e.g., Sanger sequencing, SNP arrays, or independent sequencing technologies). What additional methods were used to confirm candidate de novo variants? What fraction of potential de novo events might be missed under the current allele depth and coverage parameters?

A study that was seeking comprehensive characterization of all *de novo* mutations genome wide would undoubtedly make very different choices regarding the coverage tradeoff. A smaller study with deeper coverage would likely end up being a study that is primarily about germline mutation such as this one (Michaelson et al. 2012) where we sequenced a limited set of monozygotic twin pairs to maximize accuracy for detecting *de novo* mutations.

As is mentioned in the methods section, we use the orthogonal high-coverage short read data for the same individuals to confirm the presence of candidate *de novo* SVs. This verification involved using coverage signatures for deletions and duplications and discordant read pairs for deletions, insertions, duplications and inversions. It is unreasonable to use Sanger sequencing or other low-throughput sequencing methods to verify every single candidate *de novo* in the study.

In order to estimate the fraction of *de novo* SVs that we would potentially miss with the existing level of coverage, we looked at the coding constrained SV set (211 variants in Fig. 1C). Among 299 trios in this variant set, 47% satisfy the necessary coverage and phasing criteria required for a *de novo* SV to be detected:

1. Having 3 supporting reads in the offspring.

2. Having at least one phased reference read in each haplotype of each parent.

Among the trios which did not satisfy all these criteria, 72% did not meet condition 2, 4% did not meet condition 1, and 24% did not meet both. Based on this set of SVs we estimate that we might miss 50% of de novo SVs genome-wide.

Furthermore, in order to check if we miss any of de novo variants in the constrained coding category, we relaxed these conditions and checked every trio in this variant set with an offspring with sample quality $SQ \geq 20$ and used both SR-WGS and LR-WGS data to confirm if the variant is inherited or de novo. We did not find any new de novo variants in this category. The only de novo variant in this category was a deletion intersecting MECOM which we have already reported in Table S5 .

4. Burden Model

The description of the burden model raises questions. If single-nucleotide variants (SNVs), polygenic risk scores (PRS), and principal components (PCs) are derived from ONT data, this is problematic, as ONT is not well suited for high-quality SNV detection. Clarification is needed as to whether these inputs were instead derived from orthogonal datasets.

All SNV calls were obtained from our published short read WGS dataset on the same samples. These included de novo and missense loss of function SNVs and inherited loss of function SNVs (dnLoF, dnMIS, inhLoF), polygenic scores and ancestry principal components. The published dataset was referenced, but we have now made this more clear in the text: We tested for associations of SVs and TRs that intersect with protein-coding exons of genes in 3 categories: genes intolerant to loss-of-function ($pLI > 0.9$), genes highly expressed in fetal brain⁴³ and genes previously implicated in ASD (SFARI genes)²⁴. SV burden within the above categories was defined as the total number of intersecting rare (allele frequency < 0.05) deletions, insertions, duplications, and inversions (Table S19). Additional genome-wide SV covariates derived from LR-WGS included genome-wide mean coverage and genome-wide SV burden. Additional genetic covariates obtained from our published SR-WGS dataset⁸ included ancestry principal components (PCs; Fig. S13), burden of de novo SNV loss-of-function (dnLoF) and missense (dnMIS), inherited loss-of-function (inhLoF) variants and a polygenic risk score for autism (ASD PRS).

Additional Points for Clarification

5. Figure 1A, 1B, 1C: Please report not only absolute variant counts but also the proportion validated by orthogonal methods, as discovery rates differ by platform, but error rates may as well.

We previously provided cross-validation status for the de novo SVs. We have now added this information for all “target SVs” shown in Fig 1C in Table S3 . In this category, 90% of the SVs can be validated by SR-WGS. This involves bespoke approaches to confirm signatures of SVs in SR-WGS. Among the SVs which were only detected by LR-WGS, one was a mosaic SV depicted in Figure 2A , and the other was a complex DUP-DEL SV illustrated in Figure 3C where SR-WGS failed to detect.

A challenge that our lab and others have faced repeatedly over the years is experimental validation of variant calls detected with new technologies. Some of the reviewers can probably recall being asked to validate differentially expressed genes detected by RNASeq using qPCR! Clearly the reviewer is not asking for qPCR, but there is an analogous situation in the current study that systematic validation of all SV calls detected by LR-WGS with a 2nd (generally inferior) technology is not feasible or necessary.

6) Page 5 statement (“Structural modeling using AlphaFold 23 suggests a substantial alteration to the C-terminal structure of the mutant protein”): Please clarify the specific structural differences and their biological relevance. Likewise, in Figure 2D, clarify what difference is being

illustrated. Is there a statistical model you can apply here?

Apologies for such a dense figure. Exon 11 is tandemly duplicated and is predicted to form an in-frame tandem duplication of 66 amino acids within the STK33 protein. To make this more clear, we have now labeled each exon (11a and 11b) of the gene, and we have labeled each corresponding region of the protein structure (11a and 11b). The other parts of the protein structure are not affected by this duplication.

In order to add clarification on the confidence of the amino acid positions of the structural models we have colored the protein structures with pLDDT score in Figure S5 to show which parts of the protein are tightly packed and which parts are loose. The biological consequence of this mutation needs to be further investigated and is not in the scope of this study. To clarify the changes in the figure we have added the following text to the manuscript:

“The duplicated sequence (exon 11b, the yellow structure in Fig. 2D) is predicted to fold onto the other copy (exon 11a, the purple structure) near the C-terminal region. Further investigation is needed to predict the biological consequence of this mutation.”

Figure S5. STK33 protein structure with and without mutation colored by pLDDT score.

(A) Wild-type STK33 protein. (B) Mutant STK33 protein with exon 11 duplication.

7) **Figure 2A, 2B: What method was used to estimate copy number?**

Mosdepth was used to compute local coverage in 100 or 500 bp windows. Local GC content in the same windows is used for GC correction of the coverages. Normalization is done using coverage of the flanking regions.

A section in the methods is added to clarify these steps for copy number calculations:

“ Copy number calculation from SR-WGS

The copy number calculations in Fig. 2 and Fig. 3 are done using mosdepth₃₂ to obtain the local coverage from SR-WGS in 100 or 500 bp window sizes. Local GC content is also used for GC correction of local coverages using the loess function in R: `loess(coverage ~ GC_content)`. The

coverage values are normalized by those of the flanking regions to obtain the copy number values. "a

8) *Paper Title ("diverse functional consequences"): Please specify what these consequences are as currently, the title claim is too vague.*

We agree the title could be more specific. We have settled on "Long-Read Genome Sequencing Improves Detection and Functional Interpretation of Structural and Repeat Variants in Autism" which captures the overall results. Specifically, we demonstrate that LR-WGS substantially improves the detection of gene-disrupting structural variants and tandem repeats, particularly those at smaller scales (<1,000 bp) that are often missed by SR-based methods. LR-WGS also offers distinct advantages with respect to determining the functional consequences of SVs Long reads provide precise resolution of fine-scale structural features and complex rearrangements. Phase information facilitates detection of somatic mosaicism. Joint analysis of phased genetic variants and DNA methylation enables functional characterization of variants in FMR1 and imprinted genes.

9) *Methylation analysis of FMR1: This section seems disconnected from the main analyses. Consider moving it later in the manuscript, expanding it to include a broader methylation analysis (e.g., imprinting), or clarifying its role in the overall narrative.*

We agree that it would strengthen the paper to expand the methylation analysis. As suggested here (and in reviewer 1 comment 18b), we have combined the phased SV data with phased methylation to find signatures consistent with imprinting disorders: a loss of function variant that is on the expressed allele of an imprinted gene (Fig. 4 , Fig. S10).

Referees' reports, second round of review

Reviewer #1: The authors addressed all my comments and revised the manuscript. I look forward to reading the final version once published!

Reviewer #3: Authors have addressed my concerns.

Authors' response to the second round of review

No response